# Multi-population Black Hole Algorithm for the problem of data clustering

**Sinan Q. Salih** [1]*, **AbdulRahman A. Alsewari** [2]*, **H. A. Wahab**[3], **Mustafa K. A. Mohammed**[4], **Tarik A. Rashid** [5], **Debashish Das**[2], **Shadi S. Basurra**[2]

1 Technical College of Engineering, Al-Bayan University, Baghdad, Iraq, 2 Data Analytics & AI research Group, College of Computing and Digital Technology, Faculty of Computing Engineering and the Built Environment, Birmingham City University, Birmingham, United Kingdom, 3 Faculty of Computing, Kuantan, Malaysia, 4 University of Warith Al-Anbiyaa, Karbala, Iraq, 5 Computer Science and Engineering Department, University of Kurdistan Hewler, Erbil, Iraq

* sinan_salih@outlook.com (SQS); rahman.alsewari@bcu.ac.uk (AAA)

**Data Availability Statement:** The datasets used in this article have been collected from the machine learning repository and others can be found at these links: 1) Iris dataset: https://archive.ics.uci.

## Abstract

The retrieval of important information from a dataset requires applying a special data mining technique known as data clustering (DC). DC classifies similar objects into a groups of similar characteristics. Clustering involves grouping the data around k-cluster centres that typically are selected randomly. Recently, the issues behind DC have called for a search for an alternative solution. Recently, a nature-based optimization algorithm named Black Hole Algorithm (BHA) was developed to address the several well-known optimization problems. The BHA is a metaheuristic (population-based) that mimics the event around the natural phenomena of black holes, whereby an individual star represents the potential solutions revolving around the solution space. The original BHA algorithm showed better performance compared to other algorithms when applied to a benchmark dataset, despite its poor exploration capability. Hence, this paper presents a multi-population version of BHA as a generalization of the BHA called MBHA wherein the performance of the algorithm is not dependent on the best-found solution but a set of generated best solutions. The method formulated was subjected to testing using a set of nine widespread and popular benchmark test functions. The ensuing experimental outcomes indicated the highly precise results generated by the method compared to BHA and comparable algorithms in the study, as well as excellent robustness. Furthermore, the proposed MBHA achieved a high rate of convergence on six real datasets (collected from the UCL machine learning lab), making it suitable for DC problems. Lastly, the evaluations conclusively indicated the appropriateness of the proposed algorithm to resolve DC issues.

## 1 Introduction

The past few decades have seen various nature-inspired algorithms being highlighted to resolve numerical optimization issues. These algorithms are key players in unravelling a multitude of engineering optimization problems due to worldwide investigation and their exploitability. They are characterised by their mimicry of living organisms' behaviour in nature, like

edu/ml/datasets/Iris 2) Wine dataset: https://
archive.ics.uci.edu/ml/datasets/Wine 3) CMC
dataset: https://data.world/uci/contraceptive-
method-choice 4) Cancer dataset: https://archive.
ics.uci.edu/ml/datasets/Breast+Cancer+Wisconsin+
%28Original%29 5) Glass dataset: https://archive.
ics.uci.edu/ml/datasets/Glass+Identification 6)
Vowel dataset: https://www.isical.ac.in/~sushmita/
patterns/vowel.dat.

**Funding:** Thanks to Birmingah City University for
supporting this project. The funder provided
support in the form of salaries for authors [A.
Alsewari,D. Das, S. Basurra], but did not have any
additional role in the study design, data collection
and analysis, decision to publish, or preparation of
the manuscript. The specific roles of these authors
are articulated in the 'author contributions' section.

**Competing interests:** he authors have declared
that no competing interests exist.

the general fauna living on land or in water, respectively. As such, metaheuristic searching optimization has recently gained a huge interest in being utilised in a wide range of well known optimization problems and engineering applications. It has applications in various fields, such as power optimization [1–5]; text clustering [6]; smart traffic management [7]; robotic [8]; networking [9–11]; data security [12, 13], engineering [14–16]; and machine learning [17–25].

Regardless of the different concepts and natural inspirations behind the approaches for meta-heuristic searching optimization, they have one common fundamental structure. That is the utilisation of a heuristic-based selective search method in the solution space to find the best solution that optimizes a given objective function. In the case of a multi-optimization problem, a set of objective functions are applied as long as the set of constraints is preserved. The factors that have attracted researchers in recent times to these algorithms are the rapid advancement of hardware performance and their ability to solve numerous problems in the engineering field; they are also attractive due to the simplicity of their objective function and constraints. Several nature-inspired searching optimization frameworks have been proposed based on the multitude of natural occurrences seen worldwide; some of these natural occurrences include the particle swarms, krill herds hunting behaviour of bats, black holes, bee food searching behaviour, ant colonies, improvisation process of jazz musicians, and evolutionary algorithms such as genetic algorithm, and differential evolution [26–35].

There were several attempts for modifying and enhancing the performance of the nature-inspired algorithms by updating the architecture of the algorithms for handling different case studies. In [36] the authors introduced an improved version of the Harmony Search (HS) algorithm, called the Improved Harmony Search (IHS) algorithm. The IHS algorithm combines the power of HS with fine-tuning capabilities of mathematical techniques to achieve high-quality solutions with fewer fitness function evaluations. The authors demonstrated the effectiveness of the proposed approach in several test problems, where it outperforms other evolutionary and mathematical programming techniques reported in the literature. The work by [37] proposed a discrete variation of the Grey Wolf Optimizer (GWO) called the Discrete GWO (DGWO) for scheduling dependent tasks in cloud computing environments. The scheduling process in DGWO is formulated as a minimization problem for computation and data transmission costs. The algorithm utilizes the largest order value (LOV) method to convert the continuous candidate solutions produced by GWO to discrete candidate solutions. On the other hand, the island model is another way to enhance the nature-inspired algorithms is a commonly used technique in nature-inspired algorithms, such as genetic algorithms and Cuckoo Search Algorithm evolutionary algorithms. It involves partitioning the population into multiple sub-populations or islands and applying the optimization algorithm independently to each island [38, 39].

Data clustering (DC) refers to classifying similar objects into a group whose content differed significantly from the objects contained in another group. DC is an unsupervised learning process as the objects are placed in unspecified and predetermined clusters. In contrast, classification is a method of learning with supervision, whereby objects are classified into predetermined groups (clusters). However, clustering is associated with the issue of the absence of antecedent knowledge of the dataset provided, as well as the challenging selection of various input parameters like the number of clusters, the number of nearest neighbours, and more. Wrongly selected parameters would inevitably result in bad outcomes. Moreover, below-par precision has also bogged down the algorithms in the case of datasets that host clusters of dissimilar complex shapes, densities, sizes, noise, and outliers [40].

Various real-world applications have implemented DC methods widely. The main aim of this approach is to partition data objects such that the accumulated distances between data

objects and their respective centroids are minimized. By clustering, objects within a cluster should have as much similarity as possible while being significantly different from objects in other clusters. In other words, DC can be viewed as an optimization problem where the objective is to partition a given set of data points into a fixed number of clusters such that the within-cluster similarity is maximized and the between-cluster similarity is minimized.

Among the common approaches to solving DC, problems are to formulate it as a meta-heuristic optimization problem [41–46]. The study by [27] recently devised one of the meta-heuristic optimization methods known as a "black hole," which replicates the natural action of a black hole (BH) of drawing in neighbouring stars. The concept of BHA and its interaction with the neighbouring stars formed the basis of the BHA algorithm. In this regard, the work presented by [27] has flaws in terms of exploration as the process of obtaining an optimal resolution necessitates too many reiterations. The BHA and its enhanced versions have been utilised to tackle several well known optimization problems recently [47–61].

Recently, several metaheuristics have been enhanced by incorporating a multi-swarm or multi-population approach, including Genetic Algorithm (GA) [62], Artificial Bee Colony (ABC) [63], and Particle Swarm Optimizer (PSO) [15, 64–66], and Nomadic People Optimizer (NPO) [33] due to their capability to use different populations with their parameters set and they can simultaneously implement search space. As a result, they have significantly enhanced the performance of the original metaheuristic [67, 68]. This paper proposed multiple BHA optimization as a generalization to BHA optimization, in which the algorithm no longer depended on one best resolution. Instead, a set of best solutions were generated and called MBHA, which was maintained for some time in the search process. Furthermore, the algorithm's objective function was replaced with an objective function of higher effectiveness to resolve the clustering issue. Additionally, it was also compared with the original BHA algorithm according to several datasets.

The rest of this article is organised as follows: The Section 2 will focus on some earlier reported data clustering methods, while Sections 3 and 4 focus on BHA and the proposed MBHA, respectively. Finally, Section 5 summarizes the experimental results, while Section 6 summarises the work.

## 2 Background

This section aims to offer an overview of the data clustering optimization problem and the black hole optimization algorithm. First, the section explains data clustering as an optimization problem, providing the necessary mathematical formulation. Additionally, the section presents a review of the most significant related works. The second subsection explains the original version of Black Hole Algorithm (BHA), and discusses its advantages and drawbacks.

### 2.1 The problem of data clustering

Clustering is a crucial approach to unsupervised data classification that involves grouping a set of vectors or patterns (such as data items, observations, or feature vectors) in a multi-dimensional space [69–71]. The process of DC is determined by the dataset categorization concept using a specific number of clusters while reducing the intra-object distance within each cluster. The rearrangement of a given set of data patterns is referred to as cluster analysis; it is usually represented by one of two things: 1) a vector of measurements; or 2) a point in a multi-dimensional space. The procedure is conducted to create clusters that are differentiated by similarity attributes [72]. Some of the common application areas of DC are image processing, analysis of medical images, as well as statistical data analysis. They are also useful in various science and engineering fields and are sometimes used interchangeably with statistical data analysis. The

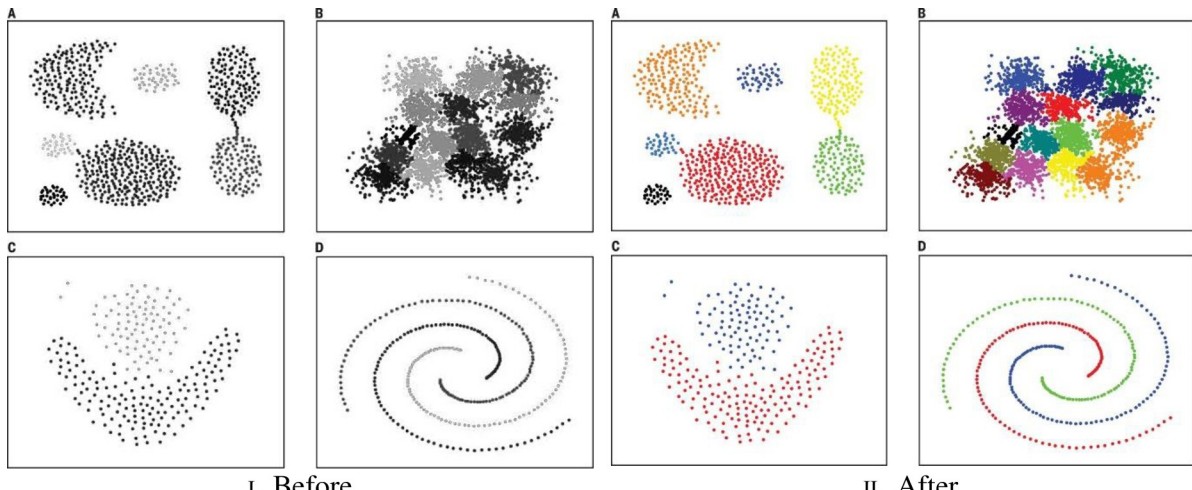

I. Before　　II. After

**Fig 1.** (I) and (II) the difference between the data before and after performing the clustering. I. Before, II. After.

differences across clusters can be attributed to their sizes, shapes, and densities, as seen in Fig 1.

Nevertheless, noise present in the data presented may pose a challenge for cluster detection, whereby the ideal cluster is fundamentally designated as "a set of points that is compact and solitary. Although humans are commonly ascribed to their cluster seeking proficiency in probably three dimensions, automatic algorithms remain as the go-to for high-dimensional data." This fact, alongside the undesignated number of clusters that are yet to be described for a provided dataset, has consistently generated thousands of clustering algorithms underlined in publications [73]. Meanwhile, the learning task can be described in pattern recognition, in which the data analysis section is commonly linked with predictive modelling. In this case, the training data is allocated to predict the unknown test data behaviour. Assessment of the data similarity may require the use of "distance measures; the problem may be designed thus: given N records data, each record is assigned to only one of the K clusters. After that, clustering is done using several criteria that serve as the process objective function (OF). The minimizing of the sum of squared ED between each record and the center of the related cluster" is one of the commonly observed features. This is shown below.

$$F(O.Z) = \sum_{i=1}^{N} \sum_{j=1}^{K} \|O_i - Z_j\|^2, \tag{1}$$

where $\|Q_i - Z_j\|$ is the Euclidian Distance (ED) between a data record $O_i$ and the cluster center $Z_j$. $N$ and $K$ are the numbers of data records and the number of clusters, respectively.

Combining a nature-inspired optimization algorithm with a clustering algorithm has led to the generation of optimal solutions. The study by [74] has displayed the method of adaptive time-dependent transporter ant for clustering (ATTA-C), which underlines alterations to the standard "Ant Colony Optimizer (ACO)" Ant-based clustering algorithm. It aims to subject high dissimilarities to a penalty, enhance the spatial separation between clusters, and facilitate clustering procedures. Achieving this requires the calculation of the fitness value for each clustering solution, which is carried out using a neighbourhood function (NF). Meanwhile, the study [75] has underlined a novel Particle Swarm Optimizer (PSO) approach for clustering issues, which is implementable in the case of a known or unknown number of clusters. The algorithm is termed CPSO and proceeds according to the *gbest* neighborhood topology,

encoding cluster centroids in particles and possibly generating new partitions during optimization. This occurs either by the removal or splitting of these clusters, until the allocated number of clusters is yielded.

Furthermore, an improved version of the Firefly Algorithm (FA) was proposed by [76] for a given dataset, in which the FA is employed and implemented for the training set to obtain the cluster centre via random selection of 75% of the dataset provided. Meanwhile, the remaining 25% dataset is termed a test dataset and utilized to investigate the FA algorithm performance [77]. The Krill Herd Algorithm (KHA) is mostly used to display a simulated herding pattern of each krill individual. The density-based approach utilized allows the discovery of clusters, subsequently undermining the region of adequately high density into clusters of krill individuals arbitrarily shaped in the climate. The objective goal of the krill movement is the minimum distance between individual krill from the food source and highly-dense herds. That is considered via foraging movement and random diffusion. In the case of a density-based cluster, it can be described as a set of density-linked objects of maximum concerning density-reach capability and noisy objects. The study [44] has previously suggested an artificial bee colony (ABC) clustering approach be subjected to categorical data. A one-step k-modes procedure is first developed for this particular approach before it is incorporated with the ABC to yield a categorical data cluster. Meanwhile, the study by [78] introduced C-ESA as a hybridization of the K-means clustering algorithm and Elephant Search Algorithm (ESA) for data clustering and obtaining the best centroid location and clustering precision enhancement.

In [79], a map/reduce programming for the ABC algorithm has been designed, capable of configuring and incorporating data in a multi-node environment. The ABC allowed the speediest completion time during execution, displaying its high efficiency for all types of data due to the parallelism attribute it offers. It also provides the amalgamation of local and global search techniques to achieve a trade-off between exploration and exploitation capabilities in obtaining optimized clusters. Similarly, the designed map/reduce programming utilizing ABC mechanisms is incorporated in a single node and multi-mode Hadoop platform, whereby the mapper phase generates the best fitness value by mimicking the behaviour of the employed bee. Meanwhile, the reducer phase achieves the probability value for cluster optimization by mimicking the onlooker and employed bees. The resulting experimental outcomes have predicted consecutively run times of varying dataset sizes in single-node and multi-node climates. Upon evaluating the performance displayed by the ABC scheme alongside the conventional Differential Evolution (DE) and PSO schemes, the ABC method was found to show superior results for optimal cluster selection compared to the remaining options. Furthermore, it also minimized the time for execution and errors in classification in the optimal cluster selection for multi-node Hadoop cluster architecture.

Meanwhile, fresh heuristic gravitational-based for data clustering has been described by [80], which answers to the excess centroid movement. Owing to the excess of centroid velocity history in the gravitational clustering algorithm, this serves as a way of improving the balance between exploration and exploitation capabilities. The technique includes an initialization phase that uses the variance and median approach so to avoid random initialisation effects. Following that, the centroid's accumulated velocity history is removed, leaving only the force of the data points in the cluster associated with the centroid to influence its position throughout any iteration.

Besides, an alternative clustering method that is effective and superior shown by [81] has opted for the application of a nature-inspired krill herd algorithm. The problem is translated into an optimization search problem via objective function minimisation to distinguish the optimum centre of each cluster. Then, multiple real and synthetic databases are reviewed, with comparison studies undertaken to elucidate the purpose of the ESKH-C technique. The

technique is specifically implemented to attain quality clustering using dissimilar dimensional real data and synthetic databases alike. The predicted outcome of confidence results from the simulation studies also indicated that the technique could group optimal cluster groups having different data shapes, sizes, dimensions, and densities. In [82], modified Bee Colony Optimization (MBCO) has been implemented, with its hybridization with k-means serving as a way of its application to data clustering. The technique is synonymous with bees' traits of forgiveness and a fair chance, which is seen for trustworthy bees or their opposite alike. It is also associated with the probability-based selection (Pbselection) approach that allocates unassigned data points in every iteration. The paper by [83] has revealed a semi-supervised K clustering framework, whereby a K-means clustering framework is initially used for the gene data. Following this, an enhanced semi-supervised K means clustering is implemented for greedy iteration to identify the K mean clustering and obtain improved outcomes. Simulations have subsequently proven that a global semi-supervised K clustering algorithm offers superior capacity for optimization and cluster effect in comparison with MDO algorithm.

Overcoming the issue of local optimum in K-Means also resulted in [84], in which a new clustering framework is designed via hybridized Crow Search Optimization (CSA). A novel population-based metaheuristic optimization algorithm is rooted in the crows' intelligent behaviour. Similarly, a K-Means clustering algorithm called CSAK means has also been suggested, whereas [43] has recently designed an Elephant Herding Optimization suited for clustering tasks. In this method, intra-cluster distance and cost function are reduced.

## 2.2 Black Hole Algorithm

Based on the black hole phenomena, the BHA is based on the core premise of an expanse of space housing a large volume of mass. The mass is concentrated within, making it impossible for any adjacent object to escape its gravitational pull. If one were to fall victim to the event, one would be obliterated from the cosmos, including light. The method is made up of two parts: 1) star movement and 2) star re-initialization upon entering the D-dimensional hypersphere around the *BH* (i.e., the event horizon). It functions as follows: the first step is the initialization of the N+1 stars, $x_i \in R^D$, $i = 1, \ldots, N+1$ in the search space, where $N$ = population size. The best value after subjection to a fitness evaluation is then recognized as the black hole $x_{BH}$. Because it is known to be static, no movement is visible until other stars reach a higher resolution. As a result, the number of individuals searching for the best value is equal to $N$, and in each generation, a star is shifting towards the *BH* as seen in the following equation [27]:

$$x_i(t+1) = x_i(t) + rand \times (x_{BH} - x_i(t))$$
$$i = 1, 2, 3, \cdots, N, \tag{2}$$

where *rand* is a random number in the range [0,1].

Furthermore, the BHA suggests that a star that comes too close to the BHA and passes through the event horizon would be removed. The following equation describes the radius of the event horizon (R) [27]:

$$R = \frac{f_{BH}}{\sum_{i=1}^{N} f_i}, \tag{3}$$

where $f_i$ and $f_{BH}$ are the *BH*'s and $i^{th}$ star's fitness values, respectively. $N$ represents the number of stars considered as the candidate solutions.

When $R$ is greater than the distance between a potential solution and the *BH* (the best solution), the related candidate is automatically collapsed, causing the formation of a new possible solution that is distributed arbitrarily over the search space. The BHA is characterized by a

simple structure requiring no parameters and can be easily implemented. Compared to the other heuristics, the BHA converges to the global optimum in all iterations, unlike the other heuristics that can be trapped in locally optimal solutions [27, 85]. Although using BHA as a clustering method is associated with outstanding results, it has drawbacks due to a lack of balance between exploration and exploitation capabilities. Finding a better solution than the existing BHA will alter the direction of a star, thereby changing the star's orientation into a new BH. Furthermore, the event horizon must be conceptualized due to the stars' possible rapid convergence for the solution space to be absorbed by the BH. This problem is caused by the lack of exploration capabilities by the BHA. It does not, however, provide intensified processes for exploration or information collection regarding previously found solutions; instead, it is just a restart approach that is applied to each star [86].

## 3 Multi-population Black Hole Algorithm

The weakness in the exploration capability of the Black Hole Algorithm (BHA) stems from its low diversity population. The algorithm tends to converge too quickly to local optima, which limits its ability to explore the search space and find global optima [87]. Therefore, in case of the exploitation capabilities are being performed more than the exploration capabilities, the chances of being trapped in a local optimum are increased. In this paper, an enhanced version of the BHA algorithm was proposed and called the "Multi-Population Black Hole (MBH) Algorithm" for the problem of data clustering. MBHA is based on the original BHA algorithm but uses multiple populations instead of a single one. Each population comprises several candidate solutions (stars) that undergo random generation in the search space. Then, the populations are initialised and each of their fitness values is assessed, whereby the best candidate having the best fitness value is chosen as the black hole. At the same time, the rest reverts to becoming normal stars. As the black hole can absorb stars around it, such a process of star absorption occurs after the black hole and stars are initialised, at which the stars move. The absorption process has been formulated as seen below:

$$x_i(t + 1) = x_i(t) + c \times rand \times (x_{BH} - x_i(t))$$
$$i = 1.2. \cdots N, \tag{4}$$

where $x_i(t+1)$ and $x_i(t)$ are the location of the i$^{th}$ star at iteration $t$ and $t+1$, $x_{BH}$ is the location of the black hole in the search space, c is a constant, $rand$ is a random number in the interval [0, 1], and $N$ is the number of stars (candidate solutions) in the population. The constant $c$ is utilized to restrict solutions scattering in the space, as well as to yield a higher convergence speed for the algorithm.

While running the algorithm, a star (or the BH) in a population may arrive at a location offering lesser cost compared to the current black hole or not reach it. This results in the concept of *Search Counter* (*SC*), which defines the number of times a population evolves without finding an improved fitness value. Therefore, if a star reaches a better location, there will be a probability of generating a new star for that population (prob$_{generating\_star}$), and this probability is formulated as follows:

$$prob_{generating\_star} = 1 - \frac{SC}{SC_{max}} \tag{5}$$

where $SC_{max}$ is the maximum value of $SC$.

After checking the probability of generating a new star, the $SC$ will be reset to zero. This probability helps the population that loses many stars due to the cessation of evolution for some time to acquire new stars and give them a longer life span. A population loses some of its

stars due to crossing the event horizon in case of the limitations of a black hole in space shaped as a sphere. The black hole will suck in every star that ventures into its event horizon, whereby every star death is characterized by a new replacement star of probability ($prob_{replace}$) that is arbitrarily distributed in the search space. The $prob_{replace}$ is formulated as $prob_{generating\_star}$, which will help the progressing population to keep its number of stars as large as possible. The calculation for the radius of the event horizon in the BHA algorithm is done using Eq (3).

A population must be omitted if the number of its stars becomes less than the minimum allowed a number of stars in a population. At each iteration, there will be a probability of generating a new population ($prob_{generating\_population}$), which will help to explore the entire search space and avoid the local minima at a minimum number of iterations (speed up the convergence to global optima in early iterations).

$$prob_{generating\_population} = \frac{rand}{number\ of\ populations}, \tag{6}$$

where *rand* is a random number in the interval [0.1]. The solutions of the new population are generated in two ways: 1) arbitrarily in the search space, and 2) arbitrarily chosen from other populations. The ratio $r_g$ is used to mix between the two ways and is formulated as follows:

$$r_g = \frac{itr}{max\ iterations}, \tag{7}$$

where *itr* is the iteration of generating the new population and max iterations refers to the total number of iterations. Therefore, the search process during the early iterations is considered to be a global search ($r_g$) is small, and the solutions are arbitrarily generated in the search space. As the iterations continue, it becomes a local search ($r_g$) is become larger and the solutions are taken from other populations. Note that the value of $r_g$ can be also selected as a constant. Thus, to generate a new population, there are two cases: if $r_g$ is less than 50% of the total number of iterations then generate a new random population, otherwise, generate the population based on the position of the global best black hole ($BH_G$) as shown in the following equations:

$$Pop\ (P) \begin{cases} r_g \leq 0.5\ then\ genearet\ a\ random\ population \\ otherwise,\ \ generate\ based\ on\ BH_G\ via\ eq\ (9) \end{cases} \tag{8}$$

$$Pop(P).X_i = Pop(P).X_i + (BH_G - Pop(r_1).X_{r_2})*rand \tag{9}$$

where $X_i$ represents a new star in the population $P$, while $r_1$ and $r_2$ represent a randomly selected population, and a randomly selected star from that population, and *rand* is a random number in the range [0,1]. This work can overcome BH's weaknesses and make a good balance between global search and local search. The key processes for the enhanced BHA algorithm are subsequently summarised using the following pseudocode in Fig 2, while the flowchart is given in Fig 3.

## 4 Results and discussion

MBHA performance was assessed by carrying out two sets of experiments. Firstly, several mathematical objective functions with multiple local minima were used to further evaluate the developed algorithm and to compare it with the original BHA and other related works. Secondly, MBHA algorithm has been validated and tested based on six benchmark datasets, and compared to other powerful state-of-art algorithms.

<u>Algorithm 1: MBHA Algorithm</u>

1.  **Inputs:** Dataset or Test Function $Num\_of\_Pops, PopSize, Max\_Sols, Min\_Sols, MaxItr, SC_{max}, Upper, Lower$
2.  **Output:** Best Solution $X_{BH}$
3.  **Procedure**
4.  <u>Define</u> Objective Function $f(x_i)$
5.  <u>Initialize</u> all populations of stars with random locations in the search space, as follows:
$$pop(p).x_i = (Upper - Lower) \times rand + Lower$$
6.  <u>Evaluate</u> the objective function for each star in each population via $f$
7.  <u>Select</u> the best star in each population as the black hole ($x_{BH}$).
8.  *__While__ $itr \leq MaxItr$*
9. *__For__ $p = 1\ to\ Num\_of\_Pops$*
10. <u>Update</u> the location of each star in the $pop(p)$ via the following equation:
$$pop(p).x_i(t+1) = pop(p).x_i(t) + c \times rand \times \big(pop(p).x_{BH} - pop(p).x_i(t)\big)$$
11. <u>Evaluate</u> the objective function for each star
12. <u>Determine</u> the best star *(best_star$_j$).*
13. *__If__ $Cost(best\_star_j) < Cost\ (pop(p).x_{BH})$ __Then__*
14. *__If__ $rand < prob_{generating\_star}$ && $Number\_Of\_Solution(pop(p)) < Max\_Sols$ __Then__*
15. $New(star) \xrightarrow{add} pop(p)$
16. *__End If__*
17. $SC = 0$
18. $pop(p).x_{BH} = best\_star_j$
19. *__Else__*
20. $SC = Min(SC + 1, SC_{max})$
21. *__End If__*
22. *<u>Calculate</u> the value of the event horizon for $pop(p)$ via the following equation:*
$$R_p = \frac{f(pop(p).x_{BH})}{\sum_{i=1}^{N} f(pop(p).x_i)}$$
23. *__For__ each star $X_i$ in the population*
24. *__If__ $X_i$ crosses the event horizon ($R_p$) __Then__*
25. <u>Remove</u> the star $X_i$
26. <u>Generate</u> a new star via Step 5
27. *__End If__*
28. *__End For__*
29. *__End For__ // **p***
30. <u>Set</u> the global best solution over all population as the global best black hole ($BH_G$)
31. *__For__ $p = 1\ to\ Num\_of\_Pops$*
32. <u>Calculate</u> the probability of generating or replacing the population, as follows:
$$prob_{generatin\_population} = \frac{rand}{no.of\ population}$$
33. *__If__ $rand < prob_{generating\_population}$ __Then__*
34. <u>Calculate</u> the generating ratio $r_g$ as follows:
$$r_g = \frac{itr}{max\ iterations}$$
35. *__If__ $r_g \leq 0.5$ __Then__*
36. <u>Generate</u> new $pop(p)$ of stars with random locations in the search space via *step* 5
37. *__Else__*
38. <u>Generate</u> new $pop(p)$ based on $BH_g$ as follows:
$$pop(p).x_i = pop(p).x_i + (BH_G - pop(r_1).x_{r2}) * rand$$
39. *__End If__*
40. *__End If__*
41. *__End For__*
42. *__Loop__*
43. Return $X_{BH}$

**Fig 2. The Pseudocode of MBHA algorithm.**

## 4.1 Evaluation on benchmark test functions

To highlight MBHA for superior exploration compared to the standard BH, further verification has been carried out via a set of multi-model types of objective functions in a multi-

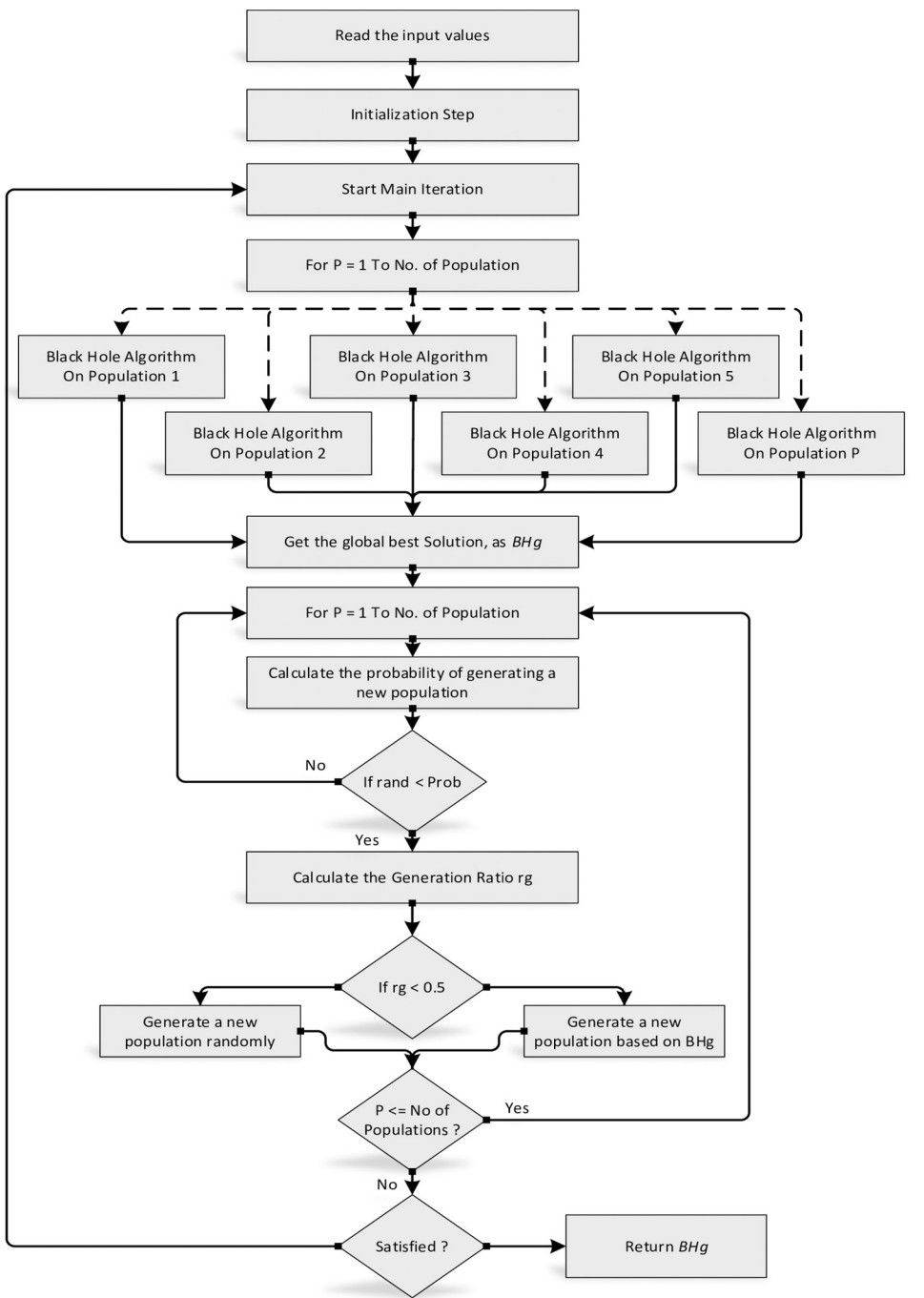

**Fig 3. The flowchart for the MBHA algorithm.**

dimensional space. Table 1 has succinctly outlined elements like the functions and their key features such as the "Name, Dimensions (D), Upper and Lower Boundaries (UB, LB)," the optimal solution (Opt) values are also stated in Table 1 while the parameter setting is in Table 2. These parameter settings were utilized in their default values as specified in the original versions. Moreover, each function is also associated with the generation of the convergence curve of the search, which is then differentiated from the actual BHA algorithm. The

**Table 1. Benchmark test functions.**

| Fun | Name | Test | D | LB | UB | Opt |
|---|---|---|---|---|---|---|
| $f_1$ | Sumsquare | $f_1(x) = \sum_{i=1}^{N} x_i^4$ | 30 | -10 | 10 | 0 |
| $f_2$ | Rastrigin | $f_2(x) = \sum_{i=1}^{N} \{x_i^2 - 10\cos(2\pi x_i) + 10\}$ | 30 | -5.12 | 5.12 | 0 |
| $f_3$ | Quartic | $f_3(x) = \sum_{i=1}^{n} ix_i^4 + random(0,1)$ | 30 | -1.28 | 1.28 | 0 |
| $f_4$ | Ackley | $f_4(x) = -20e^{-0.02\sqrt{D^{-1}\sum_{i=1}^{D} x_1^2}} - e^{D^{-1}\sum_{i=1}^{D}\cos(2\pi x_i)} + 20 + e$ | 30 | -32 | 32 | 0 |
| $f_5$ | Alpine No.1 | $f_5(x) = \sum_{i=1}^{D} |x_i \sin(x_i) + 0.1x_i|$ | 30 | -10 | 10 | 0 |
| $f_6$ | Griewank | $f_6(x) = \sum_{i=1}^{Dim} \frac{y_i^2}{4000} - \prod_{i=1}^{Dim}\cos\left(\frac{y_i}{\sqrt{i}}\right) + 1$ | 30 | -600 | 600 | 0 |
| $f_7$ | Penalized | $f_7(x) = \sum_{i=1}^{Dim-1}(y_i - 1)^2 \times (1 + sin^2)(3\pi y_{i+1}) + (y_{Dim} - 1)^2(1 + sin^2(2\pi y_{Dim})) + sin^2(3\pi y_1)$ | 30 | -50 | 50 | 0 |
| $f_8$ | Zakharov | $f_8(x) = \sum_{i=1}^{n} x_i^2 + \left(\frac{1}{2}\sum_{i=1}^{n} ix_i\right)^2 + \left(\frac{1}{2}\sum_{i=1}^{n} ix_i\right)^4$ | 30 | -5 | 10 | 0 |
| $f_9$ | Sphere | $f_9(x) = \sum_{i=1}^{N} x_1^2$ | 30 | -100 | 100 | 0 |

simulation was done using Matlab 2018a on a PC with the following specifications: Core i7, 16 GB RAM, 3.6 GHz, 64-bit Windows 10 OS.

The performance of the new MBHA was benchmarked against 9 popular metaheuristics which are Genetic Algorithm (GA) [88], Arterial Bee Colony (ABC) algorithm [89], Particle Swarm Optimization (PSO) [90], Levy Firefly Algorithm (LFFA) [91], Grey Wolf Optimizer (GWO) [92], Ant Colony Optimization Algorithm (ACO) [30], Bat algorithm (BA) [93], Flower Pollination Algorithm (FPA) [94], and Blackhole (BH) [27]. The assessments and experiments were carried out accordingly, with MBHA and BHA being subjected to 30

**Table 2. Parameter setting.**

| Method | Parameters | Value |
|---|---|---|
| General | Swarm/Colony/Population Size | 25 |
| | Iterations | 250 |
| | No. of Runs | 30 |
| FA | $\beta_0$ | 1.0 |
| | $\gamma$ | 1.0 |
| | $\alpha$ | 0.2 |
| | $\delta$ | 0.96 |
| PSO | $\omega$ | 0.742 |
| | $c_1, c_2$ | 1.42 |
| GA | Migration Fraction | 0.2 |
| | Crossover Fraction | 0.8 |
| BA | Pulse Rate ($r$) | 0.9 |
| | Min Frequency ($f_{min}$) | 0 |
| | Max Frequency ($f_{max}$) | 2 |
| | Decrease Sound Loudness ($a$) | 0.9 |
| | Weighting Value ($\delta$) | 0.9 |
| | Weighting Value ($\Phi$) | 0.1 |
| ABC | No. of Source | Size / 2 |
| | Limit | 50 |
| GWO | $a$ | (2 –> 0.1) |
| FPA | Levy flight $\lambda$ | 1.5 |
| | Switch Probability $P$ | 0.8 |

**Table 3. The results of the standard and modified Black Hole Algorithm.**

| Fun | Statistics | GA | ABC | PSO | LFFA | GWO | ACO | BA | FPA | BHA | MBHA |
|---|---|---|---|---|---|---|---|---|---|---|---|
| $f_1$ | Best | 4.14580 | 2.79E-16 | 2.13485 | 0.00774 | **0.00000** | 0.00156 | 2.267E+06 | 4.97E-04 | 3.34E-04 | **0.00000** |
| | Mean | 5.94750 | 2.72E-16 | 4.98451 | 0.21006 | **0.00000** | 0.02943 | 2.318E+06 | 0.00105 | 0.00348 | **0.00000** |
| | Std. Div | 2.13540 | 8.51E-12 | 3.94512 | 0.34752 | **0.00000** | 0.08790 | 5.125E+04 | 4.41E-04 | 3.12E-03 | **0.00000** |
| $f_2$ | Best | 2.10490 | 1.40E-11 | 0.02448 | 4.94E-10 | **0.00000** | 0.90001 | 3.91E-09 | **0.00000** | 0.00845 | **0.00000** |
| | Mean | 3.30850 | 8.83E-13 | 2.15168 | 2.06E-07 | **0.00000** | 1.00043 | 4.24528 | **0.00000** | 0.08394 | **0.00000** |
| | Std. Div | 3.54780 | 2.76E-12 | 1.07664 | 5.18E-08 | **0.00000** | 0.90536 | 3.47563 | **0.00000** | 0.01945 | **0.00000** |
| $f_3$ | Best | 3.458920 | 0.11531 | 1.3389 | 0.00409 | 0.00284 | 0.06348 | 0.10786 | 0.01741 | 0.02348 | **1.43E-04** |
| | Mean | 5.489530 | 0.19593 | 6.9606 | 0.02542 | 0.00379 | 0.08815 | 0.15314 | 0.02845 | 0.03154 | **9.15E-04** |
| | Std. Div | 0.832110 | 0.05549 | 0.6477 | 0.02312 | 0.00134 | 0.04413 | 0.00984 | 0.00148 | 0.00284 | **5.38E-04** |
| $f_4$ | Best | -199.58290 | -200 | -199.9877 | -200 | -200 | -200 | -199.9900 | -199.8449 | -199.7399 | -199.5829 |
| | Mean | -199.83310 | -200 | -199.9439 | -199.9994 | -200 | -200 | -199.9989 | -199.9545 | -199.6224 | -199.8331 |
| | Std. Div | 1.042200 | 0.000000 | 0.0371910 | 0.00013675 | 5.49E-10 | 0.5563240 | 0.04977150 | 0.0241990 | 0.0194490 | 1.04220 |
| $f_5$ | Best | 0.000640 | 0.000420 | 0.004250 | 0.000240 | 0.001160 | 0.004930 | 1.02E-040 | 5.82E-050 | 0.004810 | **4.91E-05** |
| | Mean | 1.063090 | 0.285680 | 2.675700 | 0.000290 | 0.107970 | 0.021710 | 0.336930 | 2.48E-030 | 0.087410 | **2.48E-04** |
| | Std. Div | 1.793080 | 0.624730 | 12.34900 | 0.000370 | 0.257690 | 0.009280 | 0.040300 | 0.000480 | 0.038470 | **0.000310** |
| $f_6$ | Best | **0.000000** | 4.261E-060 | 0.156760 | 3.20E-070 | **0.000000** | **0.000000** | 3.33E-090 | 0.000190 | 0.0015840 | **0.000000** |
| | Mean | **0.000000** | 0.00350 | 0.242080 | 1.51E-060 | **0.000000** | **0.000000** | 1.65E-050 | 0.000480 | 0.0096120 | **0.000000** |
| | Std. Div | **0.000000** | 0.00670 | 0.093740 | 1.88E-060 | **0.000000** | **0.000000** | 1.99E-050 | 0.000820 | 0.0841230 | **0.000000** |
| $f_7$ | Best | 0.897650 | 0.479890 | 5.523E+080 | **0.000000** | 0.137320 | 15.37690 | 0.816750 | 0.145480 | 0.122450 | **0.000000** |
| | Mean | 0.564320 | 0.449980 | 7.899E+080 | **0.000000** | 0.237520 | 32366.200 | 1.342110 | 1.164730 | 0.266400 | **0.000000** |
| | Std. Div | 0.003180 | 0.004780 | 1.439E+080 | **0.000000** | 0.056760 | 59623.510 | 0.006710 | 0.407210 | 0.057890 | **0.000000** |
| $f_8$ | Best | 1112.2050 | 7726.2470 | 3.554120 | 1021.3090 | 1337.8030 | 2214.4670 | 3.556760 | **0.000000** | 13234.2410 | **0.000000** |
| | Mean | 2673.2490 | 8094.7050 | 4.777460 | 2777.6890 | 2035.7420 | 3457.8990 | 4.787670 | **0.000000** | 4.409E+160 | **0.000000** |
| | Std. Div | 189.74560 | 246.11360 | 0.854470 | 171.73270 | 3506.2020 | 189.78670 | 0.897870 | **0.000000** | 1.5E+160 | **0.000000** |
| $f_9$ | Best | 2.124610 | 0.004320 | 1.29450 | 0.001280 | **0.000000** | 0.048710 | 0.578430 | 0.000940 | 0.017450 | **0.000000** |
| | Mean | 3.984520 | 0.006450 | 2.77070 | 0.003000 | **0.000000** | 0.066430 | 0.767410 | 0.008450 | 0.044780 | **0.000000** |
| | Std. Div | 2.648710 | 0.031840 | 1.08310 | 0.001050 | **0.000000** | 0.003840 | 0.688170 | 0.054910 | 0.006480 | **0.000000** |

different runs each. As a result, the best mean, error rate, and standard deviation were calculated, as seen in Table 3 for each algorithm.

Although the statistical results presented in Table 3 provides a first insight into the performance of MBHA, a Wilcoxon Signed-Rank Test pair-wise statistical test with a statistical significance value $\propto = 0.05$ is utilized for a better comparison. A Wilcoxon signed rank test is needed to compare the performance of MBHA against standard BHA and PSO individually.

The null hypothesis ($H_0$) for Wilcoxon signed Rank test is that there is no significant median difference between the mean pair of samples. The results are compared to other methods at a 95% level of confident. Here, if the Wilcoxon statistic is less or equal to the alpha ($\alpha = 0.05$), then $H_0$ will be rejected. To perform the statistical calculations, the SPSS statistics Software Version. In Table 4, the statistical results of BHA, PSO, GA, ABC and ACO algorithms compared to MBHA are given.

The convergence curve was also generated for the searching pattern of the 6 functions and compared with that of the original BHA. As can be seen from Figs 4 to 9 the MBHA has shown faster convergence curves. For the test problems, MBHA showed a better fitness value than BHA during all the optimization processes. This means the MBHA is more efficient than BHA and more suitable for the optimization problem.

**Table 4. The Wilcoxon signed rank test.**

| $f_n$ | MBHA vs GA | | MBHA vs ABC | | MBHA vs BH | | MBHA vs PSO | | MBHA vs ACO | |
|---|---|---|---|---|---|---|---|---|---|---|
| | P-value | Decision | P-value | Decision | P-value | Decision | P-value | Decision | P-value | Decision |
| $f_1$ | 0.033895 | Reject $H_0$ | 0.000094 | Reject $H_0$ | 0.000053 | Reject $H_0$ | 0.000067 | Reject $H_0$ | 0.000113 | Reject $H_0$ |
| $f_2$ | 0.000053 | Reject $H_0$ | 0.000065 | Reject $H_0$ | 0.000273 | Reject $H_0$ | 0.000112 | Reject $H_0$ | 0.000051 | Reject $H_0$ |
| $f_3$ | 0.000051 | Reject $H_0$ | 0.000067 | Reject $H_0$ | 0.000053 | Reject $H_0$ | 0.000066 | Reject $H_0$ | 0.000054 | Reject $H_0$ |
| $f_4$ | 0.000048 | Reject $H_0$ | 0.000049 | Reject $H_0$ | 0.000080 | Reject $H_0$ | 0.000051 | Reject $H_0$ | 0.000012 | Reject $H_0$ |
| $f_5$ | 0.000043 | Reject $H_0$ | 0.000053 | Reject $H_0$ | 0.000050 | Reject $H_0$ | 0.000070 | Reject $H_0$ | 0.000064 | Reject $H_0$ |
| $f_6$ | 0.000053 | Reject $H_0$ | 0.000065 | Reject $H_0$ | 0.000078 | Reject $H_0$ | 0.000061 | Reject $H_0$ | 0.000036 | Reject $H_0$ |
| $f_7$ | 0.000064 | Reject $H_0$ | 0.000053 | Reject $H_0$ | 0.000053 | Reject $H_0$ | 0.000012 | Reject $H_0$ | 0.000075 | Reject $H_0$ |
| $f_8$ | 0.000053 | Reject $H_0$ | 0.000043 | Reject $H_0$ | 0.000036 | Reject $H_0$ | 0.000049 | Reject $H_0$ | 0.000053 | Reject $H_0$ |
| $f_9$ | 0.000066 | Reject $H_0$ | 0.000094 | Reject $H_0$ | 0.000012 | Reject $H_0$ | 0.000059 | Reject $H_0$ | 0.000067 | Reject $H_0$ |

## 4.2 Evaluation on benchmark dataset

To ensure a fair comparison with existing methods, the same datasets used in the original version of the black hole algorithm and related works were utilized. Using different datasets would make it difficult to compare performance. Although testing on multiple datasets is important, consistency in dataset selection was prioritized. Six datasets were utilised to evaluate the performance of the suggested algorithm for data clustering: Iris, Wine, Glass, Cancer, Contraceptive Method Choice (CMC), and Vowel. Table 5 has outlined each of their specific attributes, whereby the datasets were all obtained from the UCI ML laboratory.

1. **Iris dataset.** Consisting of 150 arbitrary samples of flowers, the dataset's samples possessed four features of the iris flower and were grouped into 3 groups that were made up of 50 instances.

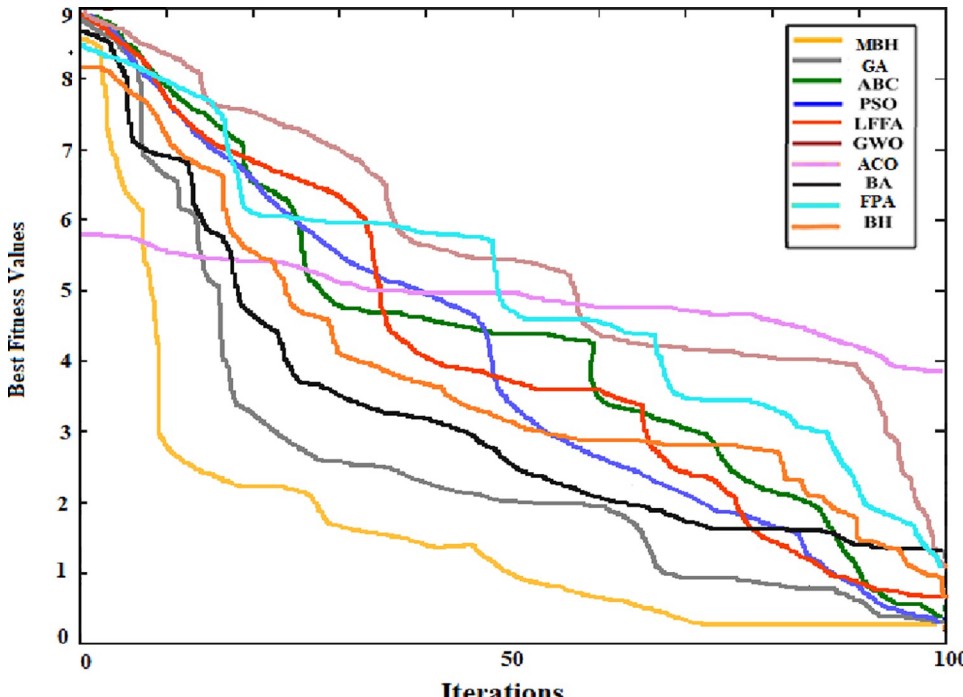

**Fig 4. The convergence of ($f_1$).**

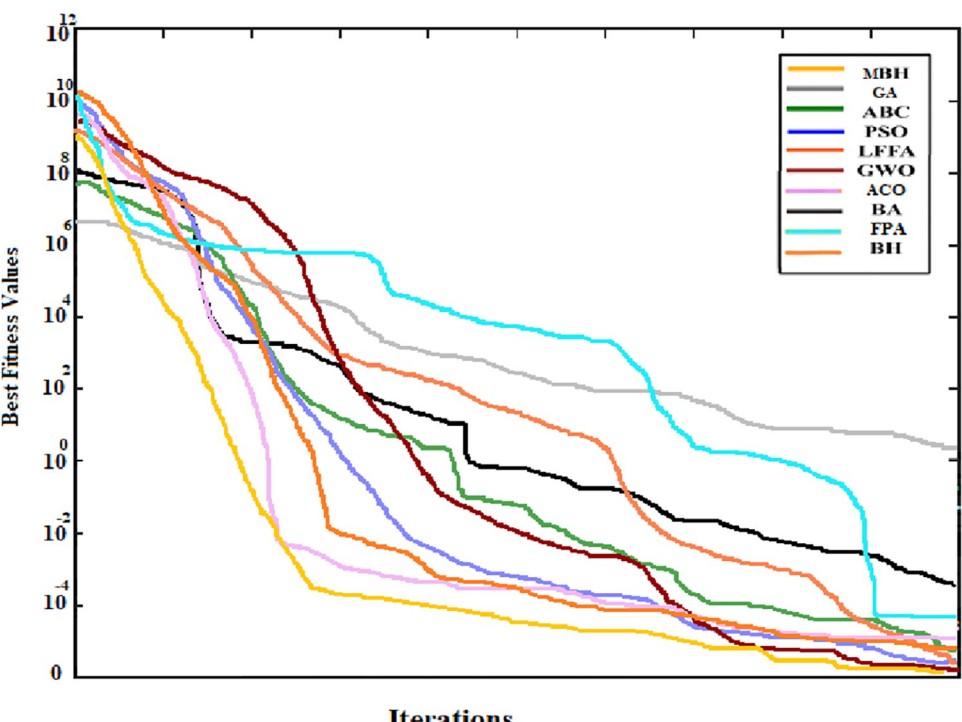

**Fig 5. The convergence of ($f_2$).**

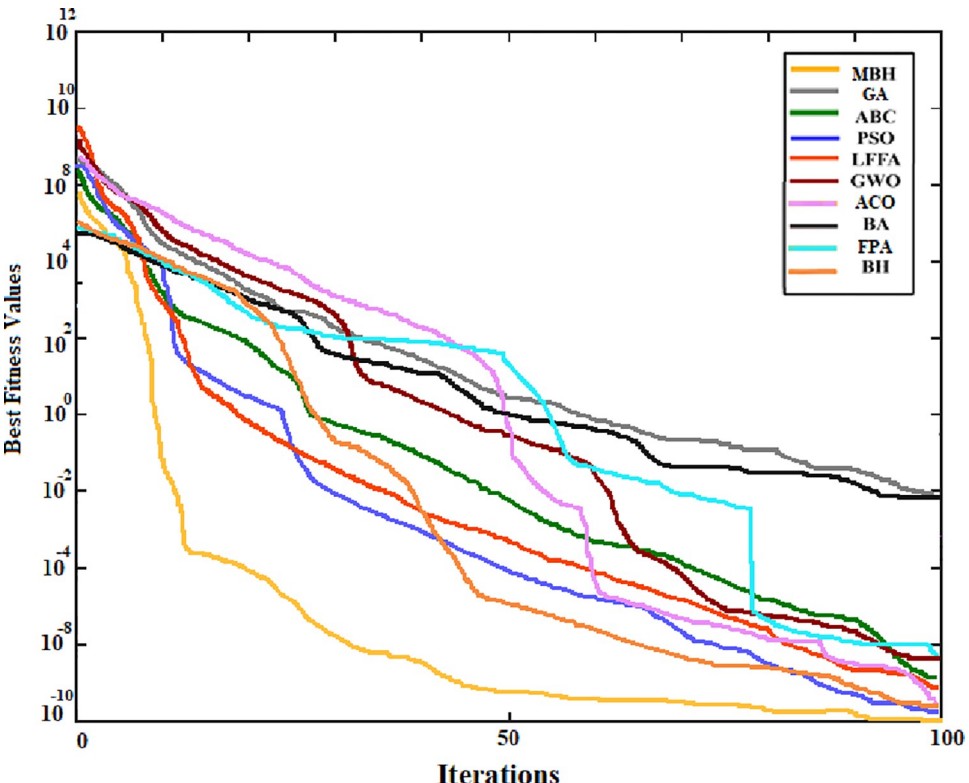

**Fig 6. The convergence of ($f_3$).**

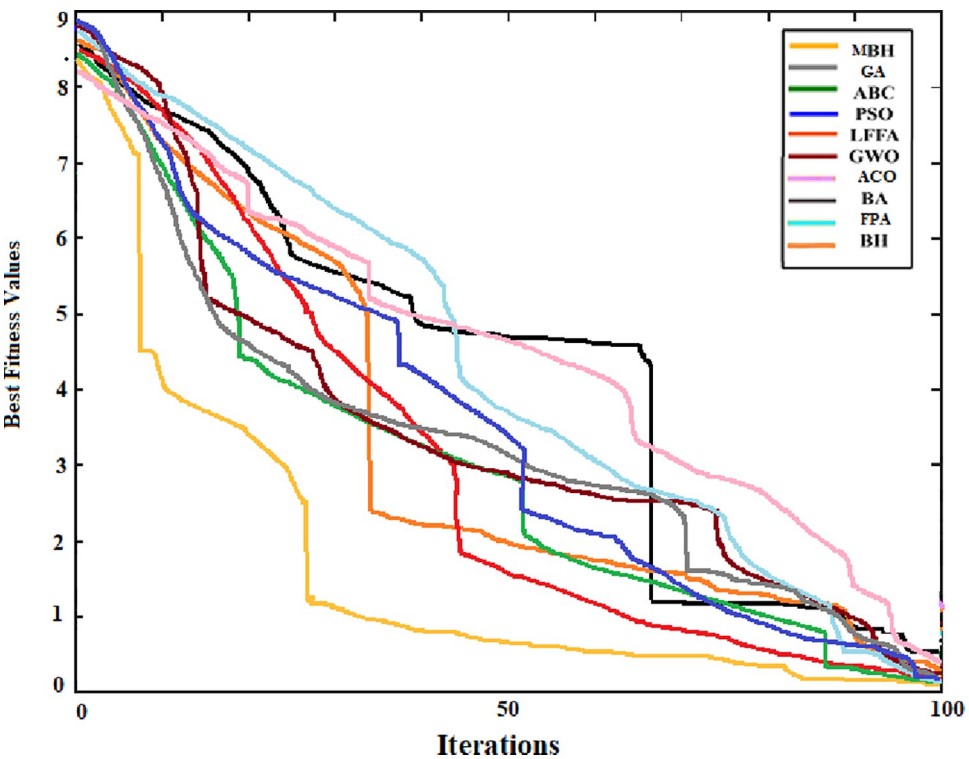

**Fig 7. The convergence of ($f_4$).**

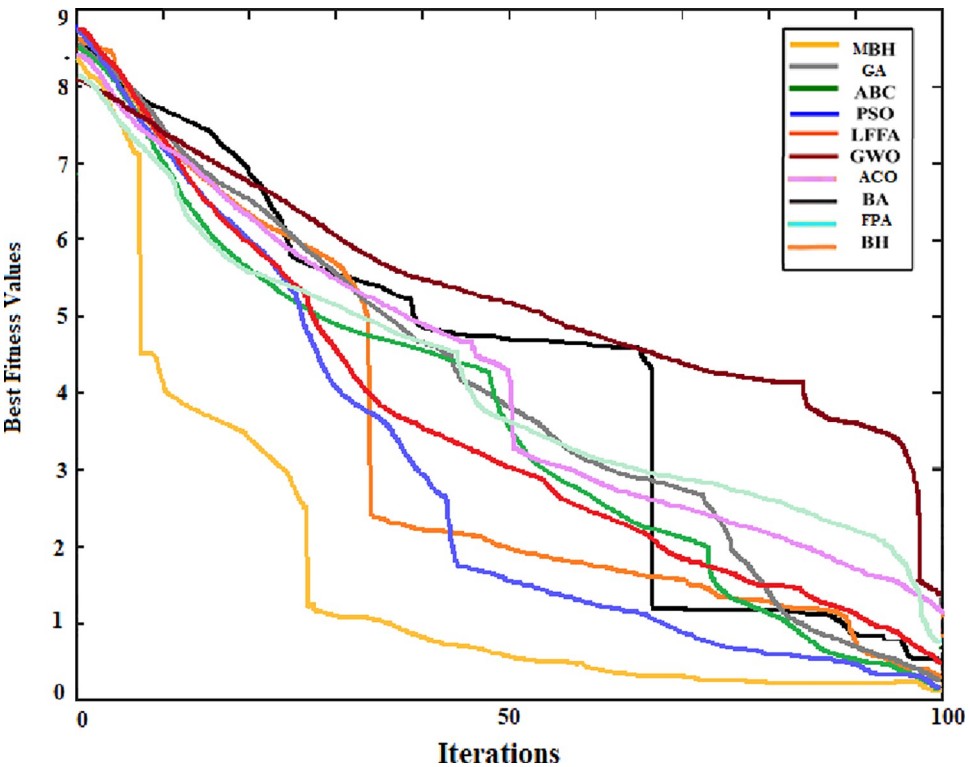

**Fig 8. The convergence of ($f_5$).**

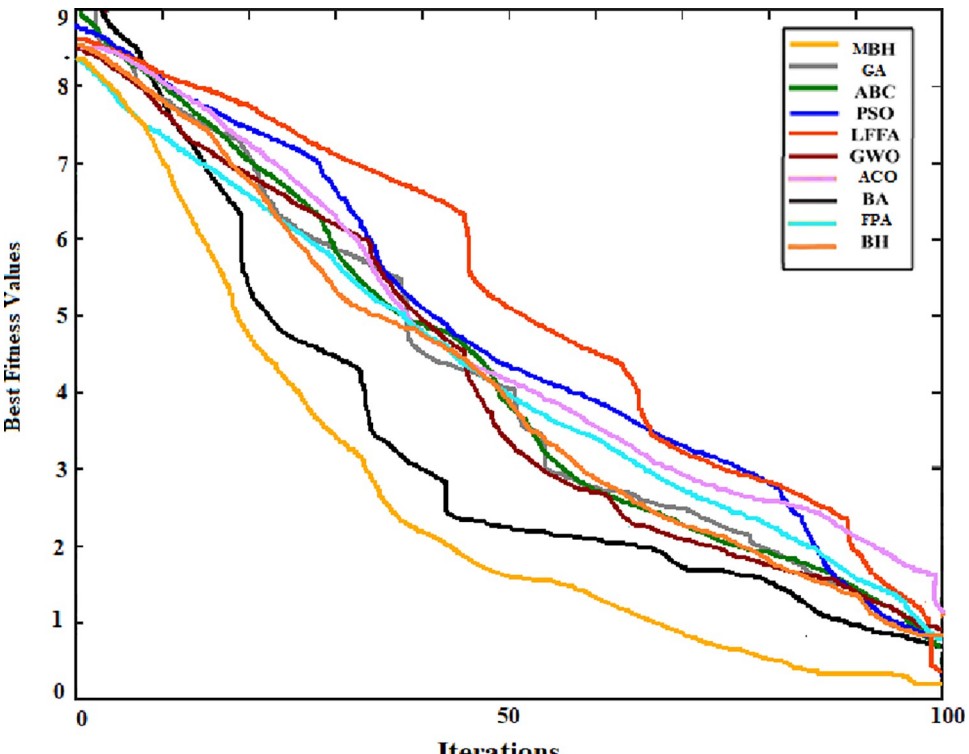

**Fig 9. The convergence of ($f_6$).**

2. **Wine dataset.** For this dataset, the quality of the wine was depicted according to its physi-cochemical attributes, as they were originally harvested from an identical Italian region but of 3 different cultivars. The three wine types were associated with 178 instances each, with 13 numeric features representing the number of 13 components in each wine type.

3. **CMC dataset.** This dataset is a subset of the 1987 National Contraceptive Prevalence Survey carried out in Indonesia. The sample size was made up of married women who, during the interview period, were either not pregnant or unknown of their pregnancy. It underlined the problem of anticipating recent choices of contraceptive techniques (i.e. no use, short-term use, or long-term use) per a woman's socioeconomic and demographic attributes.

4. **Cancer dataset.** This dataset is representative of the Wisconsin breast cancer database; it is made up of 9 components with 683 instances; the 9 components are "Clump Thickness, Cell Size Uniformity, Cell Shape Uniformity, Marginal Adhesion, Single Epithelial Cell Size, Bare Nuclei, Bland Chromatin, Normal Nuclei, and Mitoses." Every instance was attributed to being possibly benign or malignant, respectively.

**Table 5. Main characteristics of the test datasets.**

| Datasets | No. of classes | No. of features | No of instances | Size of classes |
|----------|----------------|-----------------|-----------------|-----------------|
| Iris | 3 | 4 | 150 | 50,50,50 |
| Wine | 3 | 13 | 178 | 59,71,48 |
| CMC | 3 | 9 | 1473 | 629,334,510 |
| Cancer | 2 | 9 | 683 | 444,178 |
| Glass | 6 | 9 | 214 | 70,17,76,13,9,29 |
| Vowel | 6 | 3 | 871 | 72,89,172,151,207,180 |

5. **Glass dataset.** This dataset is made up of 214 objects with 9 features that included "refractive index, silicon, potassium, sodium, calcium, magnesium, aluminium, barium, and iron." Meanwhile, six types of glass were used in the data sampling process; these are "non-float processed building windows, float processed building windows, containers, tableware, float-processed vehicle windows, and headlamps."

6. **Vowel dataset.** This dataset is made up of 871 Indian Telugu vowel sounds; the dataset also has 3 attributes that correspond to the 1st, 2nd, and 3rd vowel frequencies, as well as 6 overlapping classes.

The comparison stage was conducted by calculating four statistical values after executing the algorithms for 30 run times; the output was the sum of intra-cluster distances. These four values are (Best, Average, Worst, and Standard deviation). Additionally, all algorithms have been compared based on the value of the error rate. These two measurements can be defined as follows:

1. The sum of the distances between the clusters as a measure of internal quality: The calculation and summing up of the intra-cluster distances between the data centre and each data object is shown in Eq 1. A higher cluster quality is typically correlated to a smaller sum of intra-cluster distances, in which the sum of the distances between the clusters was one of the fitness components evaluated in this study.

2. Error Rate (ER) as an external quality measure: The equation below displays the percentage of misplaced data objects:

$$ER = \frac{Number\ of\ misplaced\ objects}{total\ number\ of\ objects\ within\ dataset} \times 100 \qquad (10)$$

Several metaheuristic methods were compared with the performance of the proposed algorithm, such as K-means [72], PSO [35], ACO [95], KH [77], GSA [41], BB-BC [41], CS [96], TS [97], and BHA [27]. In addition, MBHA was also subject to a comparison with 9 of the recently modified hybrid meta-heuristics reported in the literature; these metaheuristics included: K-means++ [98], IKH [98], BSF-ABC [99], ACPSO [66], H-KHA [81], K-MCI, MCI [100], and NM- PSO, K-NM-PSO [99]. The results of the comparison based on the standard meta-heuristics clustering frameworks and the modified hybrid meta-heuristic for a better benchmarking of the MBHA are shown in Tables 6 and 8.

A summary of error rate and intra-cluster distances is shown in Table 6. Each of the algorithms was implemented for 30 runs, and after the simulation runs, the values for the best, average, worst, standard deviation and error rate were for each algorithm. In the Table, the values in bold were the best-derived values using algorithms for each dataset. The results of the experiments showed that MBHA outperformed BHA and K-means. Further comparisons showed that the suggested technique achieved the least standard deviation compared to the other algorithms, implying that the MBHA is always at its minimum value.

Furthermore, the Iris dataset depicted MBHA algorithms having a convergence of 96.522 for each run. In contrast, the wine dataset indicated that the MBHA revealed the superior solution for worst 16,294.230, average 16,293.400, and standard deviations 0.7623. Moreover, the CMC dataset also showed the best, worst, and mean solutions obtained by MBHA of 5528.800, 5531.220, and 5530.000, with a standard deviation of 0.3466. On the other hand, the K-means, PSO, ACO, KH, GSA, BB-BC, CS, TS, and BHA failed to result in the best solutions.

**Table 6. The sum of intra-cluster distances and error rate obtained by MBHA and STANDARD algorithms on different datasets.**

| DS | Criteria | K-means | PSO | ACO | KH | GSA | BB-BC | CS | TS | BHA | MBHA |
|---|---|---|---|---|---|---|---|---|---|---|---|
| Iris | Best | 97.32590 | 96.89428 | 97.10070 | 97.43300 | 96.68794 | 96.67648 | 97.98364 | 97.36590 | 96.65589 | 96.51300 |
| | Average | 106.57660 | 97.23280 | 97.17150 | 103.03600 | 96.73105 | 96.76537 | 102.51332 | 97.86800 | 96.65681 | 96.52200 |
| | Worst | 123.96950 | 97.89733 | 97.80840 | 108.87000 | 96.82463 | 97.42865 | 106.76087 | 98.56940 | 96.66306 | 96.53200 |
| | standard | 12.938 | 0.347168 | 0.367 | 3.410 | 0.02761 | 0.20456 | 2.18224 | 0.530 | 0.00173 | 0.00010 |
| | Error rate | 13.42 | 12.58 | 10.00 | 10.78 | 10.04 | 10.05 | 09.80 | 10.74 | 10.02 | 9.27 |
| Wine | Best | 16,555.68 | 16,345.97 | 16,530.53 | 16,391.46 | 16,313.87 | 16,298.67 | 16,363.12 | 16,666.22 | 16,293.41 | 16,289.340 |
| | Average | 17,251.35 | 16,417.47 | 16,530.53 | 16,606.90 | 16,374.30 | 16,303.41 | 16,420.81 | 16,785.45 | 16,294.31 | 16,293.400 |
| | Worst | 18,294.85 | 16,562.32 | 16,530.53 | 17,160.39 | 16,428.86 | 16,310.11 | 16,525.72 | 16,837.53 | 16,300.22 | 16,294.230 |
| | standard | 874.148 | 85,497.4 | 0 | 237.740 | 34,671.22 | 2,661.98 | 45,540.86 | 52.073 | 512.70 | 0.7623 |
| | Error rate | 31.14 | 28.52 | 28.76 | 28.92 | 29.15 | 28.52 | 29.10 | 29.56 | 28.47 | 28.25 |
| CMC | Best | 5703.200 | 5700.985 | 5701.920 | 5671.526 | 5542.276 | 5534.094 | 5778.453 | 5701.920 | 5532.883 | 5528.800 |
| | Average | 5705.370 | 5820.965 | 5819.130 | 5802.144 | 5581.945 | 5574.751 | 5962.096 | 5819.130 | 5533.631 | 5530.000 |
| | Worst | 5704.570 | 5923.249 | 5912.430 | 5966.190 | 5658.762 | 5644.702 | 6205.930 | 5912.430 | 5534.777 | 5531.220 |
| | standard | 1.033 | 46.95969 | 45.6340 | 88.219 | 41.13648 | 39.43494 | 115.239 | 45.634 | 0.59940 | 0.346 |
| | Error rate | 54.48 | 54.49 | 57.68 | 56.00 | 55.67 | 54.52 | 57.18 | 56.89 | 54.39 | 53.12 |
| Cancer | Best | 2988.43000 | 2973.50000 | 2970.49 | 3021.483000 | 2965.76394 | 2964.38753 | 3089.77652 | 2970.49 | 2964.38878 | 2961.950 |
| | Average | 2988.99000 | 3050.04000 | 3046.06 | 3107.125000 | 2972.66312 | 2964.38798 | 3200.79638 | 3046.06 | 2964.39539 | 2963.900 |
| | Worst | 2999.19000 | 3318.88000 | 3242.01 | 3250.525000 | 2993.24458 | 2964.38902 | 3476.06894 | 3242.01 | **2964.45074** | 2988.430 |
| | standard | 315.145 | 110.801 | 90.500 | 77.110 | 8.918 | **0.00048** | 102.964 | 90.500 | 0.00921 | 0.0072 |
| | Error rate | 04.39 | 05.25 | 05.30 | 03.79 | 03.74 | 03.70 | 04.94 | 03.65 | 03.70 | 3.60 |
| Glass | Best | 215.730 | 270.570 | 269.72 | 232.007 | 224.984 | 223.894 | 220.125 | 269.720 | 210.515 | 208.760 |
| | Average | 218.700 | 275.710 | 273.46 | 241.916 | 233.543 | 231.230 | 225.198 | 273.460 | 211.498 | 210.971 |
| | Worst | 227.350 | 283.520 | 280.08 | 247.085 | 248.367 | 243.208 | 227.022 | 280.080 | 213.956 | 211.569 |
| | standard | 2.456 | 4.557 | 3.584 | 5.059 | 6.139 | 4.650 | 5.662 | 3.584 | 1.182 | 0.997 |
| | Error rate | 38.44 | 30.58 | 38.67 | 40.56 | 41.39 | 41.37 | 41.89 | 40.90 | 36.51 | 30.19 |
| Vowel | Best | 149,398.66 | 148,976.01 | 149,395.60 | 155,163.59 | 151,317.56 | 149,038.51 | 149,417.31 | 149,395.6 | 148,985.61 | 148,941.00 |
| | Average | 151,987.98 | 148,999.82 | 159,458.14 | 147,411.21 | 152,931.81 | 151,010.03 | 150,186.12 | 159,458.14 | 149,848.18 | 148,943.00 |
| | Worst | 162,455.69 | 149,121.18 | 165,939.82 | 160,783.94 | 155,346.69 | 153,090.44 | 150,841.40 | 165,939.82 | 153,058.98 | 148,949.00 |
| | standard | 3425.250 | 2881.346 | 3485.381 | 3001.824 | 2486.702 | 1859.323 | 1576.369 | 3485.381 | 1306.953 | 799.890 |
| | Error rate | 43.57 | 41.92 | 42.87 | 42.55 | 42.26 | 41.89 | 42.41 | 43.90 | 41.65 | 41.53 |

Besides, the Cancer dataset via MBHA algorithm resulted in the best solution of 2961.950, whereas the glass dataset obtained 208.760 as its best optimum value. Meanwhile, MBHA showed the worst value of 211.569, in comparison with the algorithms K-means, PSO, ACO, KH, GSA, BB-BC, CS, TS, and BH, attaining the worst values over 30 runs were 227.350, 283.52, 280.08, 247.085, 248.367.21, 243.208, 227.022, 280.080, 213.956, respectively, for the glass dataset.

Similarly, the MBHA also obtained the best optimum value for the vowel dataset, which was 148,941.00. Therefore, it could be conclusively stated that the MBHA algorithm had achieved the near-best value in all runs and reassured its capacity to yield superior optimal solutions, notwithstanding a small standard deviation in a minimum number of iterations.

The algorithms were further compared statistically to check for significant differences in their performances; the statistical comparison was made using the Friedman and Iman–Davenport tests. Table 7 presents the performance of the algorithms based on the employed statistical tests.

Table 8 compared the average instar-cluster distances and error rate of various clustering algorithms; MBHA yielded the best performance and conclusively revealed superior

**Table 7. The results of the statistical analysis tests.**

| Test | Value | p-value | Results |
|---|---|---|---|
| **Friedman test** | 11.9000 | 0.02481 | Rejected |
| **Iman-Davenport** | 5.16551 | 0.00334 | Rejected |

performance for all six datasets. The Iris dataset resulted in a 0.00010 standard deviation for the proposed algorithm, which was a value that was remarkably less in comparison with the remaining clustering algorithms. However, its best solution of 96.51300 and worst solution of 96.53200 was both superior compared to the remaining.

For the Wine dataset, the proposed MBHA algorithm obtained an average value of 16,293.400 outperforming the rest of the algorithms, excluding ACPSO. Meanwhile, the CMC dataset obtained exceedingly superior performance for the proposed algorithm; the worst solution of 5531.220 was relatively better compared to the remaining algorithms by a wide margin.

The Cancer dataset revealed that the proposed MBHA depicted the best solution of 2961.950 and an average solution of 2963.900. Its standard deviation was 0.0072 and

**Table 8. The sum of intra-cluster distances and error rate obtained by MBHA and modified algorithms on different datasets.**

| DS | Criteria | K-means++ | IKH | BSF-ABC | ACPSO | H-KHA | K-MCI | NM-PSO | K-NM-PSO | MCI | MBHA |
|---|---|---|---|---|---|---|---|---|---|---|---|
| Iris | Best | 97.52590 | 96.65550 | N/A | 96.66000 | 96.55400 | 96.63620 | 96.66000 | 96.66000 | 96.65540 | 96.51300 |
| | Average | 98.58170 | 96.65550 | 97.1612547 | 96.66000 | 96.52400 | 96.66640 | 100.72000 | 96.67000 | 96.65540 | 96.52200 |
| | Worst | 122.27890 | 96.65550 | N/A | N/A | 96.98900 | 96.69190 | N/A | N/A | 96.65540 | 96.53200 |
| | standard | 5.578 | 9.872 | 0.225 | 0.001 | N/A | 0.01055 | 5.82 | 0.008 | 0 | 0.00010 |
| | Error rate | 10.101 | 9.78 | 10.45 | 9.80 | 9.000 | 11.23 | 11.13 | 10.07 | 10.9 | 9.27 |
| Wine | Best | 16,555.680 | 16,292.210 | N/A | 16,292.180 | 16,350.150 | 16,293.900 | 16,292.000 | 16,292.000 | 16,295.160 | 16,289.340 |
| | Average | 16,816.550 | 16,294.300 | 16,786.04 | 16,292.310 | 16,410.140 | 16,295.600 | 16,303.000 | 16,293.000 | 16,296.510 | 16,293.400 |
| | Worst | 18,294.850 | 16,292.840 | N/A | N/A | 16,961.140 | 16,296.940 | N/A | N/A | 16,297.980 | 16,294.230 |
| | standard | 637.140 | 0.706742 | 90.660 | 0.03 | N/A | 1.002372 | 4.28 | 0.46 | 0.907 | 0.7623 |
| | Error rate | 30.54 | 28.90 | 29.80 | 28.23 | 29.650 | 28.73 | 28.48 | 28.37 | 28.98 | 28.25 |
| CMC | Best | 5703.200 | 5693.720 | N/A | 5532.190 | 5586.532 | 5699.218 | 5537.300 | 5532.400 | 5694.280 | 5528.800 |
| | Average | 5704.190 | 5693.779 | 5882.847 | 5532.200 | 5601.681 | 5705.148 | 5563.400 | 5532.700 | 5694.580 | 5530.000 |
| | Worst | 5705.370 | 5693.735 | N/A | N/A | 5666.943 | 5721.177 | N/A | N/A | 5694.890 | 5531.220 |
| | standard | 0.955 | 0.007975 | 2.200 | 0.01 | N/A | 1.268 | 30.270 | 0.230 | 0.198 | 0.346 |
| | Error rate | 54.00 | 55.90 | 57.12 | 54.38 | 53.21 | 54.47 | 54.47 | 54.38 | 55.68 | 53.12 |
| Cancer | Best | 2986.960 | 2964.387 | N/A | 2964.390 | 2975.191 | 2962.420 | 2965.590 | 2964.500 | 2964.400 | 2961.950 |
| | Average | 2988.430 | 2964.393 | N/A | 2964.420 | 2982.437 | 3022.810 | 2977.700 | 2964.700 | 2964.410 | 2963.900 |
| | Worst | 2987.990 | 2964.389 | N/A | N/A | 2990.493 | 3150.150 | N/A | N/A | 2964.430 | 2988.430 |
| | standard | 0.689 | 0.001258 | N/A | 0.03 | N/A | 0.396 | 13.73 | 0.15 | 0.007 | 0.0072 |
| | Error rate | 3.95 | 3.69 | N/A | 3.51 | 3.86 | 4.27 | 4.28 | 3.66 | 3.78 | 3.60 |
| Glass | Best | 215.360 | 210.2520 | N/A | N/A | 213.105 | 199.860 | N/A | N/A | 213.030 | 208.760 |
| | Average | 217.560 | 222.8000 | 290.877 | N/A | 215.665 | 202.410 | N/A | N/A | 214.080 | 210.971 |
| | Worst | 223.710 | 215.9355 | N/A | N/A | 217.355 | 209.770 | N/A | N/A | 215.620 | 211.569 |
| | standard | 2.455 | 2.737 | 13.731 | N/A | 1.832 | 0.26 | N/A | N/A | 0.923 | 0.997 |
| | Error rate | 45.123 | 33.90 | 32.56 | N/A | 32.242 | 32.61 | N/A | N/A | 30.89 | 30.19 |
| Vowel | Best | 149,394.00 | 148,967.00 | N/A | 148,970.00 | 149,123.00 | 149,201.00 | 149,240.00 | 149,005.00 | 148,985.00 | 148,941.00 |
| | Average | 151,445.00 | 158,600.00 | 156,343.15 | 149,051.00 | 149,565.00 | 161,431.00 | 151,983.00 | 149,141.00 | 149,039.00 | 148,943.00 |
| | Worst | 161,845.00 | 150,172.00 | N/A | N/A | 149,999.00 | 165,804.00 | N/A | N/A | 149,102.00 | 148,949.00 |
| | standard | 3119.751 | 1732.451 | 971.037 | 67.270 | N/A | 2746.041 | 4386.430 | 120.380 | 43.735 | 799.890 |
| | Error rate | 15.364 | 41.56 | 41.90 | 41.69 | 40.10 | 41.98 | 41.96 | 41.94 | 43.0 | 41.53 |

reassuringly superior compared to K-means++, IKH, BSF-ABC, ACPSO, H-KHA, K-MCI, NM-PSO, K-NM-PSO, and MCI. In contrast, the Glass dataset obtained the best solution of 199.860 using the K-MCI algorithm, while the final dataset of Vowel yielded 148,943.00 of the best average solution by MBH. Hence, this conclusively highlighted the effectiveness of MBHA to resolve complex optimization problems, simply due to the best results generated by almost all of the datasets and upon comparison with the remaining comparative algorithms. The outcomes were specifically achieved by adding the element of new operators.

## 5 Conclusion

Black Hole Algorithm (BHA) is a newly developed optimization method that offers a promising solution for addressing complex global optimization problems. However, one of the limitations of the BHA algorithm is that the lack of balancing between the exploration and exploitation, which increases the chances of trapping in local minima, thereby preventing it from finding the optimal solution. To overcome this issue, an enhanced version of the BHA based on a new multi-population architecture, has been employed in this work by applying effective enhancements including a global exploration operator that facilitates the rapid convergence of the algorithm towards optimal solutions. The proposed algorithm is called "Multi-Population Black Hole Algorithm (MBHA)".

Simulation results demonstrate that the proposed algorithm is able to significantly reduce computation time and achieve its set objectives, thereby prompting further evaluation on data clustering problems. Furthermore, the outcomes confirm the suitability of the proposed algorithm for resolving clustering problems as compared with previous reports. Despite the numerous advantages of the MBHA algorithm, several aspects require further elucidation and investigation in future research. Firstly, the algorithm was only benchmarked on nine test functions, thus necessitating the use of more benchmark problems to provide a comprehensive assessment of its capabilities. Secondly, the issue of number of populations and their sizes presents a fascinating research area that deserves in-depth exploration. Lastly, improving the convergence of the MBHA algorithm represents a crucial research topic that warrants further investigation.

In conclusion, the proposed MBHA represents an effective optimization method that offers a viable alternative for solving complex global optimization problems. Nevertheless, further research is necessary to investigate the ability of the algorithm to handle different hard optimization problems, such as, feature selection, hyperparameters tuning for Support Vector Machine (SVM), and training artificial neural networks (ANN).

## Acknowledgments

Authors would like to thank Data Ana

## Author Contributions

**Conceptualization:** Sinan Q. Salih, H. A. Wahab.

**Data curation:** Mustafa K. A. Mohammed.

**Formal analysis:** H. A. Wahab, Tarik A. Rashid, Debashish Das.

**Investigation:** Shadi S. Basurra.

**Methodology:** Sinan Q. Salih, H. A. Wahab, Mustafa K. A. Mohammed.

**Software:** Sinan Q. Salih.

**Supervision:** AbdulRahman A. Alsewari, Shadi S. Basurra.

**Validation:** Sinan Q. Salih.

**Visualization:** Mustafa K. A. Mohammed, Tarik A. Rashid.

**Writing – original draft:** Sinan Q. Salih, H. A. Wahab.

**Writing – review & editing:** AbdulRahman A. Alsewari, Shadi S. Basurra.

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
