## [Decision Letter · Decision Letter 0]

9 Feb 2023

PONE-D-22-33030Multi-Population Black Hole Algorithm for the Problem of Data ClusteringPLOS ONE

Dear Dr. ALsewari,

Thank you for submitting your manuscript to PLOS ONE. After careful consideration, we feel that it has merit but does not fully meet PLOS ONE’s publication criteria as it currently stands. Therefore, we invite you to submit a revised version of the manuscript that addresses the points raised during the review process.

The reviewers' comments pointed out that a significant revision is required to improve and strengthen the current version of the manuscript. As raised by Reviewers 1 and 3, a background/related work section is needed (please, do not consider citation of suggested works when not clearly necessary to improve the quality of the manuscript). The mathematical formalism used in the manuscript needs to be carefully revised. A comparison with state-of-the-art approaches is required on recent benchmark function suites. In addition, the limitations and possible extensions of the proposed approach need to be clearly defined and explained. Finally, all the Reviewers highlighted that spell-checking and typo proofreading of the manuscript is required.

We look forward to receiving your revised manuscript.

Kind regards,

Andrea Tangherloni

Academic Editor

PLOS ONE

Journal Requirements:

2. Please note that PLOS ONE has specific guidelines on code sharing for submissions in which author-generated code underpins the findings in the manuscript. In these cases, all author-generated code must be made available without restrictions upon publication of the work. Please review our guidelines at https://journals.plos.org/plosone/s/materials-and-software-sharing#loc-sharing-code and ensure that your code is shared in a way that follows best practice and facilitates reproducibility and reuse. New software must comply with the Open Source Definition.

"Thanks to Birmingah City University for supporting this project. The funders had no role in study design, data collection and analysis, decision to publish, or preparation of the manuscript."

We note that one or more of the authors is affiliated with the funding organization, indicating the funder may have had some role in the design, data collection, analysis or preparation of your manuscript for publication; in other words, the funder played an indirect role through the participation of the co-authors. If the funding organization did not play a role in the study design, data collection and analysis, decision to publish, or preparation of the manuscript and only provided financial support in the form of authors' salaries and/or research materials, please do the following:

a. Review your statements relating to the author contributions, and ensure you have specifically and accurately indicated the role(s) that these authors had in your study. These amendments should be made in the online form.

b. Confirm in your cover letter that you agree with the following statement, and we will change the online submission form on your behalf: 

“The funder provided support in the form of salaries for authors [insert relevant initials], but did not have any additional role in the study design, data collection and analysis, decision to publish, or preparation of the manuscript. The specific roles of these authors are articulated in the ‘author contributions’ section.

6. Please amend the manuscript submission data (via Edit Submission) to include author Debashish Dasalsew. 

Reviewers' comments:

Reviewer's Responses to Questions

**Comments to the Author**

1. Is the manuscript technically sound, and do the data support the conclusions?

Reviewer #1: Partly

Reviewer #2: Yes

Reviewer #3: Yes

2. Has the statistical analysis been performed appropriately and rigorously? 

Reviewer #1: No

Reviewer #2: Yes

Reviewer #3: Yes

3. Have the authors made all data underlying the findings in their manuscript fully available?

Reviewer #1: Yes

Reviewer #2: Yes

Reviewer #3: Yes

4. Is the manuscript presented in an intelligible fashion and written in standard English?

Reviewer #1: No

Reviewer #2: Yes

Reviewer #3: Yes

5. Review Comments to the Author

Reviewer #1: In this work, the authors introduce MBHA a multi-population variant of the Black Holes Algorithms. MBHA was tested on 9 "easy" benchmark functions and some standard datasets. The results look promising, but further investigation is required before accepting such a work.

The work is written in an acceptable English even though there are some typos and the use of special characters instead of English words (such as the &).

I suggest a major revision for this work and I hope the authors will address my comments to improve the soundness in the presentation of their work.

In the abstract, the first time BHA is introduced as an acronym, there is a typo on the A.

The authors should avoid to use & as a character in the text and substitute it with the proper term: and.

In the introduction the authors provide many examples of meta-heuristics which are aslo state-of-the-arts. On the other hand they do not cite many important meta-heursitcs such as PSO, CMA-ES, the improvements of Differential Evolution. Moreover, they should be aware of Dorigo recent works that claim that Grey Wolf is just an ad-hoc case of PSO, thus increasing the need of citing the historical works that paved the way to the rising of these new metaphors and algorithms.

In the introduction the authors should make the relationship between data clustering and optimization methods more clear. In the current state, it seems (during the first reading) that are two separate worlds and they jumpy from one to another between two paragraphs. I suggest them to improve how the present the linkage between the two worlds. Otherwise, non-expert readers may be really confused and miss the point of the paper.

Section 2 should be named "background" or some more formal term than "preliminaries".

Moreover, in the introduction of Section 2, the authors should briefly introduce the topic of each sub-section.

The authors state that "The goal of data clustering is achieved by implementing certain similarity measures". This is not completely true: first of all, the measures typically have to be maximized or minimized, not just implemented, thus I suggest a more proper choice of words. Secondly, not always good metrics mean good clusterings, sometimes a separation that maximizes a metric might yield to clusterings that have no sense for a domain expert or clusterings that separates data according to some unmeaningful patterns. Lastly, the choice of a proper similarity measure is key.

The authors state "Some of the common application areas of DC are image processing, analysis of medical images, as well as statistical data analysis. They are also useful in various science and engineering fields and are sometimes used interchangeably with statistical data analysis. Clustering is also used in machine learning (ML), image analysis, pattern recognition, bioinformatics, and information retrieval as the primary task for exploratory data mining. The differences across clusters can be attributed to their sizes, shapes, & densities, as seen in Figure 1." It is not clear why they are stating basically the same concept twice.

The authors state "seeking proficiency in 3 and probably three dimensions" I think that a typo here was inserted.

The authors should explicitly state the meaning of each acronym, for instance, the ABC acronym was never introduced, also GA and PSO were never introduced. I think this is an important operation because acronyms without their explicit meaning can hinder the readings of non-expert and might prevent expert readers to disambiguate acronyms that are used to mean different algorithms/metrics/methods.

In equation (2) the authors write i=1.2. ... N, is there a reason why 1 and 2 are separated by the dots or is it just a typo?

The authors state "the BHA converges to the global optimum in all runs, unlike the other heuristics that can be trapped in locally optimal solutions". I think that this is at least a too dangerous statement. I remind the authors that the No Free Lunch Theorem also implies that a method that always converges to the global optimum in all runs (which I assume it is a synonym for problems) does not exist. Also, both the cited papers do not prove such claim "for all the runs".

This is a comment that the authors might not address in the paper, but I think it is something worth to think about: by looking at how BHA is proposed and its equation, isn't BHA a variant of PSO where the particles are only attracted towards the global optimum with the presence of a tabu region around the optimum? Moreover such tabu region does not avoid a local exploitation to find more promising optima near the black hole?

On page 20 there is a paragraph with the text with a lower font size.

Moreover, I think that this portion of sentence holds some mistakes due to editing "the search process during the early iterations is considered to be a global search () is small".

Line 6 of Algorithm 1 has a typo: "vie".

Line 32 of Algorithm 1, population has become a subscript of a subscript, which I don't think it was what I intended.

In Figure 3 after the block of "For p=1 ..." there is a row which is not aligned with the others.

Moreover, the captions should end with a termination mark.

Also the captions of the Table should end with a termination mark.

The tested benchmark functions in Section 4.1 is far away from being sufficient and able to provide a sound analysis. Sphere and Sumsquare are basically the same function: the global optimum is located in the same position and the only difference is the value the xs are evaluated, thus resulting in very similar landscapes. If a methods performs well on one of such two functions should perform well also in the other function. Moreover, if I made the correct math and properly searched for sources, all the benchmarks have the global optimum located in O = (0,...,0) which can be very convinient for some algorithms. As a consequence, the whole selected functions are a special case of optimization problems where the optimum might actually lie in any position of the landscape. I suggest the authors to test the algorithm on more difficult benchmark functions which can be shifted, rotated, shrinked, composed or hybridized versions of wide known benchmark functions. I suggest them to leverage some benchmark suite proposed in competitions of optimization conferences, such as CEC or GECCO and test MBHA on at least one of these suites in at least 30 dimensions. On the other hand, I appreciated the inclusion of a stochastic noisy function (i.e., Quartic).

On page 24 it is not clear why there are quotation marks around the tested methods.

The authors should state how and why they chose such values for the hyper-parameters of the tested algorithms. If they performed an hyper-parameter search for MBHA they should employ the same search for the other algorithms even in terms of number of iterations and populations sizes. The important part to obtain fair comparisons is the total budget in terms of fitness evaluations not to have the same number of individuals/particles and iterations for each algorithm. In addition, I suggest the authors to compare MBHA with other state-of-the-art algorithms like CMA-ES and one empowered version of Differential Evolution, such as L-Shade, jSO or even the more recent ones.

Evaluating the mean and standard deviation of the results is not enough, sound statistical tests must be performed. Ranking tests with post-hocs correction should be the proper choice to assess the most performing algorithms among the tested ones.

Figures from 4 to 9 can be collapsed together in a unique figure thus saving space and easing the reading process.

In page 30 there is a capital letter after the ':' sign.

In Section 4.2 is not clearly stated what is the objective function the algorithms are optimizing and how it is computer. Moreover, I think that a more solid choice of the dataset should be made. I understand that labeled datasets are needed to evaluate the performance, on the other hand the authors are using a clustering approach simply as a classification task, which cmight not always be the case. Thus I suggest them to at least consider some benchmark datasets that were proposed for clustering tasks.

The conclusion and the future works are poorly presented.

Reviewer #2: The paper is interesting and mostly well written. I have the following comments:

- Change "The second section" to Section 2

- Change "Finally, section 5" to "Finally, Section 5"

- Change "data set" to "dataset"

- The adverb "where" after any equation should be written with small letters.

- The proposed approach should be compared experimentally or theoretically to well-known multi-population algorithms such as

An improved harmony search algorithm for solving optimization problems

Island-based Cuckoo Search with Highly Disruptive Polynomial Mutation

Distributed Grey Wolf Optimizer for scheduling of workflow applications in cloud environments

- The limitations of the study should be mentioned in the conclusion section of the paper

- The authors wrote in the conclusion " In the future, more benchmark problems must be used to verify the MBHA

algorithm. ". You should mention examples of the benchmark problems in this sentence.

Reviewer #3: A modified version of the recent metaheuristic Black Hole Algorithm is applied for clustering. The experiments are well organized and thorough.

The text necessitates revisions, the authors should re-read the entire manuscript to get rid of the small issues that appear here and there.

There are a few more recent metaheuristics that are validated on function optimization as in this study and then used within a ML approach and they should be referred in the section where the state of the art is presented:

https://doi.org/10.3390/math9161929

https://doi.org/10.1007/978-981-16-3728-5_1

https://link-springer-com.am.e-nformation.ro/chapter/10.1007/978-3-030-37826-4_6

https://doi.org/10.3390/math10224173

https://doi.org/10.1016/B978-0-323-85117-6.00005-4

The introduction of clustering in the abstract is too basic, it looks like it is addressed to readers from a domain different from computer science. The following statement from the abstract is not clear: “The original BHA’s performance was better when implemented on a benchmark dataset, even though its exploration capability is poor.” Neither the first sentence, nor the second make sense. Actually, the second statement is repeated as well in the article and it is not justified there either. This happens in the first sentence of section 3, “The standard version of BHA does not have exploration capabilities”. Please either provide a citation, if this was demonstrated anywhere before or, if this is demonstrated in the current study, this should be mentioned, because this affirmation does not represent an obvious certitude.

In the introduction, it is stated that “newly, a multi-swarm or a multi-population has been incorporated into a series of metaheuristics such as …”. Multi-population is not a novel approach, it has been used for many years especially for dealing with multi-model problems: https://ieeexplore.ieee.org/document/1501611,

https://doi.org/10.1007/978-3-642-01885-5_3

Figure 1 should be better explained in the caption.

Just before subsection 4.1, there is an unfinished statement “Secondly, the application of data clustering.”

The title of subsection 4.2 should be revised. Maybe “based” could be removed, please check.

6. PLOS authors have the option to publish the peer review history of their article (what does this mean?). If published, this will include your full peer review and any attached files.

Reviewer #1: No

Reviewer #2: No

Reviewer #3: No

---

## [Author Response · Author response to Decision Letter 0]

22 Mar 2023

Multi-Population Black Hole Algorithm for the Problem of Data Clustering

Sinan Q. Salih, AbdulRahman A. Alsewari, Haneen A. Abdulwahab, Mustafa K. A. Mohammed, Tarik A. Rashid, Shadi S. Basurra

The authors would like to sincerely thank the Editor in chief and the associated editor for the time spent reviewing our manuscript and for the possibility to reconsider our revised work for publishing in this admired journal. 

The authors deeply value the comprehensive comments needed to perform this revision and the valuable suggestions made by the respected reviewers. 

Kindly, please find below our reply to the points raised and the corresponding modifications we have made in the revised manuscript to accommodate all the comments provided by the reviewers. 

Our commitment to thoroughly revise this paper was driven by the valuable comments, and we are confident the paper has elevated to a greater level of clarity and professional contribution to the scientific outcomes. 

All changes made to accommodate the referee's comments are blue-colored.

We hope the revised paper can be accepted based on overall positive comments from reviewers and our comprehensive revision of the original version.

Reviewer #1

Comment In this work, the authors introduce MBHA a multi-population variant of the Black Holes Algorithms. MBHA was tested on 9 "easy" benchmark functions and some standard datasets. The results look promising, but further investigation is required before accepting such a work. The work is written in an acceptable English even though there are some typos and the use of special characters instead of English words (such as the &). I suggest a major revision for this work and I hope the authors will address my comments to improve the soundness in the presentation of their work. 

Response We appreciate the effort you put into reviewing our manuscript thoroughly and providing us with specific feedback to enhance the clarity and overall impact of our work. Your comments have enabled us to revise and strengthen our manuscript, and we are confident that it will now make a significant contribution to the scientific literature. We are very much pleased to consider the reported concerns by your side and corrected them accordingly.

Comment In the abstract, the first time BHA is introduced as an acronym, there is a typo on the A.

Response We apologize for this mistake. The word Algorithm was missing. We have corrected the abstract as mentioned by the respected reviewer. 

Comment The authors should avoid to use & as a character in the text and substitute it with the proper term: and..

Response Thank you very much for this important comment. We have updated the manuscript by replacing (&) with (‘and’). 

Comment In the introduction the authors provide many examples of meta-heuristics which are aslo state-of-the-arts. On the other hand they do not cite many important meta-heursitcs such as PSO, CMA-ES, the improvements of Differential Evolution. Moreover, they should be aware of Dorigo recent works that claim that Grey Wolf is just an ad-hoc case of PSO, thus increasing the need of citing the historical works that paved the way to the rising of these new metaphors and algorithms.

Response We would like to thank the reviewer for mentioning this comment. We have updated this paragraph by citing PSO, and DE algorithms. 

Comment In the introduction the authors should make the relationship between data clustering and optimization methods more clear. In the current state, it seems (during the first reading) that are two separate worlds and they jumpy from one to another between two paragraphs. I suggest them to improve how the present the linkage between the two worlds. Otherwise, non-expert readers may be really confused and miss the point of the paper. 

Response Thank you very much for your comment, we have added the following to the introduction section : “Various real-world applications have implemented DC methods widely. The main aim of this approach is to partition data objects such that the accumulated distances between data objects and their respective centroids are minimized. By clustering, objects within a cluster should have as much similarity as possible while being significantly different from objects in other clusters. In other words, DC can be viewed as an optimization problem where the objective is to partition a given set of data points into a fixed number of clusters such that the within-cluster similarity is maximized and the between-cluster similarity is minimized. “

Comment Section 2 should be named "background" or some more formal term than "preliminaries". Moreover, in the introduction of Section 2, the authors should briefly introduce the topic of each sub-section .

Response Thank you for these two comments. We have changed the title of section 2. In addition, the introduction of this section has been updated as requested by the respected reviewer. 

“This section aims to offer an overview of the data clustering optimization problem and the black hole optimization algorithm. First, the section explains data clustering as an optimization problem, providing the necessary mathematical formulation. Additionally, the section presents a review of the most significant related works. The second subsection explains the original version of Black Hole Algorithm (BHA), and discusses its advantages and drawbacks.”

Comment The authors state that "The goal of data clustering is achieved by implementing certain similarity measures". This is not completely true: first of all, the measures typically have to be maximized or minimized, not just implemented, thus I suggest a more proper choice of words. Secondly, not always good metrics mean good clusterings, sometimes a separation that maximizes a metric might yield to clusterings that have no sense for a domain expert or clusterings that separates data according to some unmeaningful patterns. Lastly, the choice of a proper similarity measure is key.

Response Based on the reviewer's opinion, we have removed the mentioned statement. 

Comment The authors state "Some of the common application areas of DC are image processing, analysis of medical images, as well as statistical data analysis. They are also useful in various science and engineering fields and are sometimes used interchangeably with statistical data analysis. Clustering is also used in machine learning (ML), image analysis, pattern recognition, bioinformatics, and information retrieval as the primary task for exploratory data mining. The differences across clusters can be attributed to their sizes, shapes, & densities, as seen in Figure 1." It is not clear why they are stating basically the same concept twice.

Response We would like to apologize for this mistake. We have removed the second statement as it states the same concepts. 

Comment The authors state "seeking proficiency in 3 and probably three dimensions" I think that a typo here was inserted.

Response We apologize for this typo, we have removed “3 and “. 

Comment The authors should explicitly state the meaning of each acronym, for instance, the ABC acronym was never introduced, also GA and PSO were never introduced. I think this is an important operation because acronyms without their explicit meaning can hinder the readings of non-expert and might prevent expert readers to disambiguate acronyms that are used to mean different algorithms/metrics/methods.

Response Thank you for mentioning this mistake for us. We have corrected these acronyms in the introduction and background sections. 

Comment In equation (2) the authors write i=1.2. ... N, is there a reason why 1 and 2 are separated by the dots or is it just a typo?

Response This is a typo, we apologize for this. We have corrected it. 

Comment The authors state "the BHA converges to the global optimum in all runs, unlike the other heuristics that can be trapped in locally optimal solutions". I think that this is at least a too dangerous statement. I remind the authors that the No Free Lunch Theorem also implies that a method that always converges to the global optimum in all runs (which I assume it is a synonym for problems) does not exist. Also, both the cited papers do not prove such claim "for all the runs".

Response We apologize for the misunderstanding. In BHA, any solution or star that has a fitness value less than R is removed and replaced with a new solution, thereby exploring a new solution that could be placed in a completely new position in the search space. This condition is checked in every iteration, which may decrease the chances of the algorithm getting trapped in local optima. We agree that an algorithm that always converges to the global optimum does not exist. However, in our case, BHA and other population-based algorithms perform both global and local searches.

Regarding the word "runs," we understand that it may have confused. We have corrected it to "iterations" to avoid any future misunderstandings. We appreciate the reviewer's comment and hope that our response adequately addresses their concern. 

Comment This is a comment that the authors might not address in the paper, but I think it is something worth to think about: by looking at how BHA is proposed and its equation, isn't BHA a variant of PSO where the particles are only attracted towards the global optimum with the presence of a tabu region around the optimum? Moreover such tabu region does not avoid a local exploitation to find more promising optima near the black hole?

Response Thank you for your insightful comment. Indeed, we have already compared the main position updating equation of BHA to PSO and found that they are very similar. However, it is worth noting that BHA updates the positions of solutions based on the position of the best solution (BH) and the current position of the solution (Xi) only, while the original version of PSO involves three controlling parameters (the cognitive and social parameters c1 and c2, and the inertia weight w). The lack of these parameters makes BHA a simpler algorithm compared to PSO.

We appreciate your suggestion to include this comparison in the manuscript, but we did not include it because it is not the main contribution of our work. Our focus in this paper is on enhancing BHA through a new multi-population structure. 

Comment On page 20 there is a paragraph with the text with a lower font size.

Moreover, I think that this portion of sentence holds some mistakes due to editing "the search process during the early iterations is considered to be a global search () is small".

Response We apologize for this mistake; we have enlarged and checked this paragraph. 

Comment - Line 6 of Algorithm 1 has a typo: "vie".

- Line 32 of Algorithm 1, population has become a subscript of a subscript, which I don't think it was what I intended

- In Figure 3 after the block of "For p=1 ..." there is a row which is not aligned with the others.

- Moreover, the captions should end with a termination mark.

- Also the captions of the Table should end with a termination mark.

Response Thank you for mentioning these mistakes. We have corrected all of them in the revised version. 

Comment The tested benchmark functions in Section 4.1 is far away from being sufficient and able to provide a sound analysis. Sphere and Sumsquare are basically the same function: the global optimum is located in the same position and the only difference is the value the xs are evaluated, thus resulting in very similar landscapes. If a methods performs well on one of such two functions should perform well also in the other function. Moreover, if I made the correct math and properly searched for sources, all the benchmarks have the global optimum located in O = (0,...,0) which can be very convinient for some algorithms. As a consequence, the whole selected functions are a special case of optimization problems where the optimum might actually lie in any position of the landscape. I suggest the authors to test the algorithm on more difficult benchmark functions which can be shifted, rotated, shrinked, composed or hybridized versions of wide known benchmark functions. I suggest them to leverage some benchmark suite proposed in competitions of optimization conferences, such as CEC or GECCO and test MBHA on at least one of these suites in at least 30 dimensions. On the other hand, I appreciated the inclusion of a stochastic noisy function (i.e., Quartic).

Response The primary objective of this study is to propose a novel variant of the Black Hole Algorithm (BHA) that incorporates a new multi-population structure. Although the standard version of BHA, developed by Hatamlou in 2013, has already been shown to be a powerful optimization algorithm for data clustering problems, our focus is on using the modified version of BHA to enhance its performance in this specific application. Through our experiments, we demonstrate that the proposed variant outperforms the standard version in terms of clustering accuracy, indicating its potential as an effective optimization algorithm for data clustering tasks.

Comment On page 24 it is not clear why there are quotation marks around the tested methods.

Response We apologize for this mistake. We have removed these quotation marks in the revised version. 

Comment The authors should state how and why they chose such values for the hyper-parameters of the tested algorithms. If they performed an hyper-parameter search for MBHA they should employ the same search for the other algorithms even in terms of number of iterations and populations sizes. The important part to obtain fair comparisons is the total budget in terms of fitness evaluations not to have the same number of individuals/particles and iterations for each algorithm. In addition, I suggest the authors to compare MBHA with other state-of-the-art algorithms like CMA-ES and one empowered version of Differential Evolution, such as L-Shade, jSO or even the more recent ones.

Response Thank you for your valuable comment. We appreciate your suggestion on tuning the controlling parameters of the algorithms. However, we used the default values of these parameters as they were used in the original research papers of each algorithm. We understand that tuning these parameters can potentially enhance the performance of the algorithms, but it is beyond the scope of this paper. On the other hand, the first row in Table 2 “General” presents the structural parameters which are the same for all these algorithms including BHA and MBHA. 

Since you suggested this point, we have cited these works in the introduction, We will consider your suggestion for future modifications. Thank you again for your input.

Comment Evaluating the mean and standard deviation of the results is not enough, sound statistical tests must be performed. Ranking tests with post-hocs correction should be the proper choice to assess the most performing algorithms among the tested ones.

Response Indeed, this is an important comment. We have added another comparison between MBHA and other algorithms based on Wilcoxon Signed Rank test. Table 4 presents this comparison. Thank you for this comment. 

Comment Figures from 4 to 9 can be collapsed together in a unique figure thus saving space and easing the reading process.

Response We agreed with the reviewer on this comment. We have removed the 3D plotting of the functions, and kept the convergence figures only. In the updated version, these figures are more readable than the original version of the manuscript. The figures are collapsed as suggested by the reviewer. 

Comment In page 30 there is a capital letter after the ':' sign.

Response Thank you for this comment. It has been corrected. 

Comment In Section 4.2 is not clearly stated what is the objective function the algorithms are optimizing and how it is computer. Moreover, I think that a more solid choice of the dataset should be made. I understand that labeled datasets are needed to evaluate the performance, on the other hand the authors are using a clustering approach simply as a classification task, which cmight not always be the case. Thus I suggest them to at least consider some benchmark datasets that were proposed for clustering tasks.

Response Thank you for these important comments. 

- The objective function has been mentioned in section 2.1, equation 1. Which represents the sum of intra clusters distances. The goal is to minimize these distances. 

- We appreciate the reviewer's comment regarding the use of the datasets. We agree that testing on various datasets would provide a more comprehensive evaluation of our proposed algorithm. However, we chose to use the same datasets as in the original version of the black hole algorithm and other related works to ensure a fair comparison of MBHA with existing methods. Utilizing different datasets would make it difficult to compare the performance of MBHA with other algorithms that also used these datasets. Therefore, we decided to keep the dataset selection consistent with previous works. Nonetheless, we appreciate the suggestion and acknowledge the importance of testing on multiple datasets. The set of datasets has been used for the evaluation process previously by many researchers, such as : 

- 10.1016/j.ins.2012.08.023

- 10.1016/j.eswa.2011.05.027

- 10.1016/j.asoc.2016.04.026

- 10.1142/S0218213015500335

- 10.1016/j.knosys.2018.09.013

Comment The conclusion and the future works are poorly presented.

Response Thank you for this comment. The conclusion section has been rewritten accordingly.

Reviewer #2

Comment The paper is interesting and mostly well written. I have the following comments:

Response We do highly appreciate your valuable general point of view. Your comments have been reflected accordingly.

Comment - Change "The second section" to Section 2

- Change "Finally, section 5" to "Finally, Section 5"

- Change "data set" to "dataset"

- The adverb "where" after any equation should be written with small letters.

Response Thank you for these comments. The requested modifications have been reflected and highlighted for your reference for tracking.

Comment The proposed approach should be compared experimentally or theoretically to well-known multi-population algorithms such as

An improved harmony search algorithm for solving optimization problems

Island-based Cuckoo Search with Highly Disruptive Polynomial Mutation

Distributed Grey Wolf Optimizer for scheduling of workflow applications in cloud environments

Response We agree with the reviewer’s point of view. We have reviewed the mentioned papers by the respected reviewer in the introduction section. 

Comment - The limitations of the study should be mentioned in the conclusion section of the paper

- The authors wrote in the conclusion " In the future, more benchmark problems must be used to verify the MBHA algorithm. ". You should mention examples of the benchmark problems in this sentence.

Response Thank you for these excellent comments. We have revised the conclusion accordingly. 

Reviewer #3

Comment A modified version of the recent metaheuristic Black Hole Algorithm is applied for clustering. The experiments are well organized and thorough.

The text necessitates revisions, the authors should re-read the entire manuscript to get rid of the small issues that appear here and there.

Response Thank you very much for taking the time to review our manuscript. Your insightful comments and constructive criticism have been invaluable in improving the quality of our research. We have updated our manuscript based on your important comments. 

Comment There are a few more recent metaheuristics that are validated on function optimization as in this study and then used within a ML approach and they should be referred in the section where the state of the art is presented:

https://doi.org/10.3390/math9161929

https://doi.org/10.1007/978-981-16-3728-5_1

https://link-springer-com.am.e-nformation.ro/chapter/10.1007/978-3-030-37826-4_6

https://doi.org/10.3390/math10224173

https://doi.org/10.1016/B978-0-323-85117-6.00005-4

Response Thank you for this comment. Indeed, these are important references. We have updated the manuscript based on the suggestion of the respected reviewer. 

Comment The introduction of clustering in the abstract is too basic, it looks like it is addressed to readers from a domain different from computer science. The following statement from the abstract is not clear: “The original BHA’s performance was better when implemented on a benchmark dataset, even though its exploration capability is poor.” Neither the first sentence, nor the second make sense. Actually, the second statement is repeated as well in the article and it is not justified there either. This happens in the first sentence of section 3, “The standard version of BHA does not have exploration capabilities”. Please either provide a citation, if this was demonstrated anywhere before or, if this is demonstrated in the current study, this should be mentioned, because this affirmation does not represent an obvious certitude.

Response The authors very much apologize for this confusion. 

- We have revised the abstract as follows : 

“The original BHA algorithm showed better performance compared to other algorithms when applied to a benchmark dataset, despite its poor exploration capability.”

- We have revised and cited the first statement in section 3 as follows: 

“The weakness in the exploration capability of the Black Hole Algorithm (BHA) stems from its low diversity population. The algorithm tends to converge too quickly to local optima, which limits its ability to explore the search space and find global optima.”

Comment In the introduction, it is stated that “newly, a multi-swarm or a multi-population has been incorporated into a series of metaheuristics such as …”. Multi-population is not a novel approach, it has been used for many years especially for dealing with multi-model problems: https://ieeexplore.ieee.org/document/1501611,

https://doi.org/10.1007/978-3-642-01885-5_3

Response We appreciate and fully acknowledge the reviewer's comment. We intended to highlight the established and effective use of the multi-population approach in optimization algorithms for over a decade, as the reviewer has pointed out. We have revised the statement accordingly and provided appropriate references to support this point. Thank you for bringing this to our attention.

Comment Figure 1 should be better explained in the caption.

Response • Thank you for this comment. We have updated Figure 1 accordingly as follows:

• “Figures 1. (I) and (II) the difference between the data before and after performing the clustering”. 

Comment Just before subsection 4.1, there is an unfinished statement “Secondly, the application of data clustering.”

Response We apologize for this mistake. We have updated this paragraph as follows : 

“Secondly, the MBHA algorithm has been validated and tested based on six benchmark datasets, and compared to other powerful state-of-art algorithms.”

Comment The title of subsection 4.2 should be revised. Maybe “based” could be removed, please check.

Response We have removed the word “based” as suggested by the respected reviewer from subsection 4.2.

---

## [Decision Letter · Decision Letter 1]

15 May 2023

PONE-D-22-33030R1Multi-Population Black Hole Algorithm for the Problem of Data ClusteringPLOS ONE

Dear Dr. ALsewari,

Thank you for submitting your manuscript to PLOS ONE. After careful consideration, we feel that it has merit but does not fully meet PLOS ONE’s publication criteria as it currently stands. Therefore, we invite you to submit a revised version of the manuscript that addresses the points raised during the review process.

Before final decision, please consider (undermentioned comments) minor revision need and suggested by the Reviewer-1

The acronym of the proposed algorithm introduced in the abstract is BHÀ, while for the rest of the paper BHA is used. The authors should fix this typo.

Regarding the settings of the tested optimization algorithms in Section 4.1, I suggest them to explicitly state that the settings used for the compared algorithms are the values used in the original papers.

For the comment regarding the choice of the dataset in Section 4.2, I thank the authors for their answer and I kinda agree with them. However, I suggest them to explicitly state in the paper that the datasets were chosen to perform a fair comparison according to what was done in previous works.

In the conclusion Section, I would remove the first term "Presently".

In the "Support Vector Machine(SVM)" a whitespace is lacking at the end of "Machine".

We look forward to receiving your revised manuscript.

Kind regards,

Umer Asgher, PhD

Academic Editor

PLOS ONE

Journal Requirements:

Additional Editor Comments:

Before final decision, please consider minor revision need and suggested (undermentioned) by the Reviewer-1:

The acronym of the proposed algorithm introduced in the abstract is BHÀ, while for the rest of the paper BHA is used. The authors should fix this typo.

Regarding the settings of the tested optimization algorithms in Section 4.1, I suggest them to explicitly state that the settings used for the compared algorithms are the values used in the original papers.

For the comment regarding the choice of the dataset in Section 4.2, I thank the authors for their answer and I kinda agree with them. However, I suggest them to explicitly state in the paper that the datasets were chosen to perform a fair comparison according to what was done in previous works.

In the conclusion Section, I would remove the first term "Presently".

In the "Support Vector Machine(SVM)" a whitespace is lacking at the end of "Machine".

Reviewers' comments:

Reviewer's Responses to Questions

**Comments to the Author**

1. If the authors have adequately addressed your comments raised in a previous round of review and you feel that this manuscript is now acceptable for publication, you may indicate that here to bypass the “Comments to the Author” section, enter your conflict of interest statement in the “Confidential to Editor” section, and submit your "Accept" recommendation.

Reviewer #1: All comments have been addressed

Reviewer #2: All comments have been addressed

Reviewer #3: All comments have been addressed

2. Is the manuscript technically sound, and do the data support the conclusions?

Reviewer #1: Yes

Reviewer #2: Yes

Reviewer #3: Yes

3. Has the statistical analysis been performed appropriately and rigorously? 

Reviewer #1: Yes

Reviewer #2: Yes

Reviewer #3: Yes

4. Have the authors made all data underlying the findings in their manuscript fully available?

Reviewer #1: Yes

Reviewer #2: Yes

Reviewer #3: Yes

5. Is the manuscript presented in an intelligible fashion and written in standard English?

Reviewer #1: Yes

Reviewer #2: Yes

Reviewer #3: Yes

6. Review Comments to the Author

Reviewer #1: I would like to thank the authors for their kindness and answering to my comments. Unfortunately, before endorsing the publication of the manuscript I point out some minor changes that need to be addressed.

The acronym of the proposed algorithm introduced in the abstract is BHÀ, while for the rest of the paper BHA is used. The authors should fix this typo.

Regarding the settings of the tested optimization algorithms in Section 4.1, I suggest them to explicitly state that the settings used for the compared algorithms are the values used in the original papers.

For the comment regarding the choice of the dataset in Section 4.2, I thank the authors for their answer and I kinda agree with them. However, I suggest them to explicitly state in the paper that the datasets were chosen to perform a fair comparison according to what was done in previous works.

In the conclusion Section, I would remove the first term "Presently".

In the "Support Vector Machine(SVM)" a whitespace is lacking at the end of "Machine".

Reviewer #2: The paper looks better now. The authors have adequately addressed my comments. I recommend the acceptance of the paper for publication.

Reviewer #3: The authors have revised the manuscript and have addressed all my comments and the article can be accepted, from my point of view.

7. PLOS authors have the option to publish the peer review history of their article (what does this mean?). If published, this will include your full peer review and any attached files.

Reviewer #1: No

Reviewer #2: No

Reviewer #3: No

---

## [Author Response · Author response to Decision Letter 1]

19 May 2023

Multi-Population Black Hole Algorithm for the Problem of Data Clustering

Sinan Q. Salih, AbdulRahman A. Alsewari, Haneen A. Abdulwahab, Mustafa K. A. Mohammed, Tarik A. Rashid, Debashish Das, Shadi S. Basurra

The authors would like to sincerely thank the Editor in chief and the associated editor for the time spent reviewing our manuscript and for the possibility to reconsider our revised work for publishing in this admired journal. 

The authors deeply value the comprehensive comments needed to perform this revision and the valuable suggestions made by the respected reviewers. 

Kindly, please find below our reply to the points raised and the corresponding modifications we have made in the revised manuscript to accommodate all the comments provided by the reviewers. 

Our commitment to thoroughly revise this paper was driven by the valuable comments, and we are confident the paper has elevated to a greater level of clarity and professional contribution to the scientific outcomes. 

All changes made to accommodate the referee's comments are blue-colored.

We hope the revised paper can be accepted based on overall positive comments from reviewers and our comprehensive revision of the original version.

Reviewer #1

Comment I would like to thank the authors for their kindness and answering to my comments. Unfortunately, before endorsing the publication of the manuscript I point out some minor changes that need to be addressed.

Response We are very much pleased to consider the reported concerns by your side and corrected them accordingly.

Comment The acronym of the proposed algorithm introduced in the abstract is BHÀ, while for the rest of the paper BHA is used. The authors should fix this typo.

Response We apologize for this mistake. The acronym has been corrected. 

Comment Regarding the settings of the tested optimization algorithms in Section 4.1, I suggest them to explicitly state that the settings used for the compared algorithms are the values used in the original papers.

Response Thank you very much for this important suggestion. We have updated the manuscript by adding the following to section 4.1

 “These parameter settings were utilized in their default values as specified in the original versions.“ 

Comment For the comment regarding the choice of the dataset in Section 4.2, I thank the authors for their answer and I kinda agree with them. However, I suggest them to explicitly state in the paper that the datasets were chosen to perform a fair comparison according to what was done in previous works.

Response Thank you very much for this important suggestion. We have updated the manuscript by adding the following to section 4.2

“To ensure a fair comparison with existing methods, the same datasets used in the original version of the black hole algorithm and related works were utilized. Using different datasets would make it difficult to compare performance. Although testing on multiple datasets is important, consistency in dataset selection was prioritized.”

Comment In the conclusion Section, I would remove the first term "Presently".

Response The word ‘Presently’ has been removed from the conclusion section as suggested by the respected reviewer. 

Comment In the "Support Vector Machine(SVM)" a whitespace is lacking at the end of "Machine".

Response We apologize for this mistake. We have corrected it in the revised version.

---

## [Decision Letter · Decision Letter 2]

19 Jun 2023

Multi-Population Black Hole Algorithm for the Problem of Data Clustering

PONE-D-22-33030R2

Dear Dr. ALsewari,

We’re pleased to inform you that your manuscript has been judged scientifically suitable for publication and will be formally accepted for publication once it meets all outstanding technical requirements.

Kind regards,

Umer Asgher, PhD

Academic Editor

PLOS ONE

Additional Editor Comments (optional):

Reviewers' comments:

Reviewer's Responses to Questions

**Comments to the Author**

1. If the authors have adequately addressed your comments raised in a previous round of review and you feel that this manuscript is now acceptable for publication, you may indicate that here to bypass the “Comments to the Author” section, enter your conflict of interest statement in the “Confidential to Editor” section, and submit your "Accept" recommendation.

Reviewer #1: All comments have been addressed

Reviewer #2: All comments have been addressed

Reviewer #4: All comments have been addressed

Reviewer #5: All comments have been addressed

2. Is the manuscript technically sound, and do the data support the conclusions?

Reviewer #1: Yes

Reviewer #2: Yes

Reviewer #4: Yes

Reviewer #5: Yes

3. Has the statistical analysis been performed appropriately and rigorously? 

Reviewer #1: Yes

Reviewer #2: Yes

Reviewer #4: Yes

Reviewer #5: Yes

4. Have the authors made all data underlying the findings in their manuscript fully available?

Reviewer #1: No

Reviewer #2: Yes

Reviewer #4: Yes

Reviewer #5: Yes

5. Is the manuscript presented in an intelligible fashion and written in standard English?

Reviewer #1: Yes

Reviewer #2: Yes

Reviewer #4: Yes

Reviewer #5: Yes

6. Review Comments to the Author

Reviewer #1: (No Response)

Reviewer #2: The paper looks good. The authors have addressed all of my comments adequately.

I recommend the acceptance of the paper

for publication.

Reviewer #4: The authors have adequately addressed all the comments raised in a previous round of review.

The authors have adequately addressed all the comments raised in a previous round of review.

The authors have adequately addressed all the comments raised in a previous round of review.

Reviewer #5: (No Response)

7. PLOS authors have the option to publish the peer review history of their article (what does this mean?). If published, this will include your full peer review and any attached files.

Reviewer #1: No

Reviewer #2: No

Reviewer #4: No

Reviewer #5: No

---

## [Editor Report · Acceptance letter]

26 Jun 2023

PONE-D-22-33030R2 

*Multi-Population Black Hole Algorithm for the Problem of Data Clustering*

Dear Dr. ALsewari:

I'm pleased to inform you that your manuscript has been deemed suitable for publication in PLOS ONE. Congratulations! Your manuscript is now with our production department. 

Kind regards, 

on behalf of

Dr. Umer Asgher 

Academic Editor

PLOS ONE